# On Contraction of Sequential and Offset Rademacher Complexities

**Adam Block** [1]  **Alexander Rakhlin** [2]  **Mark Sellke** [3]

## Abstract

The Rademacher complexity of a function class is among the most basic notions of its "size" and yields classical offline generalization bounds for Lipschitz loss functions that lead in turn to a modern understanding of statistical learning. More recently, the *sequential* and *offset* Rademacher complexities were introduced to prove analogous generalization bounds for online learning and for prediction with squared loss. A fundamental structural result in the theory of Rademacher complexity, with many applications to learning theory, is the Ledoux–Talagrand contraction lemma, which states that the Rademacher complexity of a composition of a function class with a fixed Lipschitz function is at most that of the original class. We show that, under structural assumptions on the function class, this contraction extends to sequential and offset Rademacher complexity at the price of polylogarithmic factors. We further show that these logarithmic factors cannot be removed in general and, absent these additional structural assumptions, no such contraction inequality can hold. These results together indicate that the sequential and offset Rademacher complexities behave fundamentally differently from the classical Rademacher complexity with respect to contraction, which in turn has broad implications for understanding the sample complexities of online learning and regression with squared loss for composed function classes.

## 1. Introduction

In classical supervised learning, a learner is given access to $n$ samples consisting of covariates $x$ and labels $y$, with the goal of predicting future labels as well as possible relative to some fixed function class $\mathcal{F}$, which captures the prior knowledge the learner has about the problem at hand. One of the primary objectives of learning theory is to understand how many samples $n$ are required in order to learn effectively in terms of some notion of complexity, or 'size,' of the function class $\mathcal{F}$. Among the most foundational measures of size is the *Rademacher complexity* (Giné & Zinn, 1984), which measures how well a function class is able to correlate with random noise, with many classical works in learning theory precisely determining the sample complexity of learning in terms of this quantity (Bartlett & Mendelson, 2002; Bousquet, 2002; Wainwright, 2019). Even as the Rademacher complexity has been shown to govern the difficulty of classification in a broad sense, two related measures of size, the *sequential Rademacher complexity* (Rakhlin et al., 2010; 2015) and the *offset Rademacher complexity* (Liang et al., 2015; Rakhlin & Sridharan, 2014), have been shown to be central for other learning problems, such as *online learning* and *regression* (both online and offline). While the structural properties of Rademacher complexity are fairly well-understood (Bartlett & Mendelson, 2002), similar understanding for these related notions of complexity has received far less attention.

In particular, the classical Ledoux–Talagrand contraction inequality (see e.g. (Ledoux & Talagrand, 1991)) states that for any function class $\mathcal{F}$ and Lipschitz function $\ell : \mathbb{R} \to \mathbb{R}$, the Rademacher complexity $\mathcal{R}_n$ obeys

$$\mathcal{R}_n(\ell \circ \mathcal{F}) \leq \mathcal{R}_n(\mathcal{F}). \tag{1}$$

This inequality is fundamentally important in determining the sample complexity of classical learning, as it allows the general machinery of uniform convergence to be applied to learning with arbitrary Lipschitz loss functions by providing control of the size of uniform deviations of the loss class $\ell \circ \mathcal{F}$ by those of the base class $\mathcal{F}$, which is often substantially easier to analyze (Bartlett & Mendelson, 2002). Furthermore, many important function classes in learning theory can be decomposed as a composition of Lipschitz functions with simple function classes and contraction allows for easy analysis thereof.

While the contraction inequality applies to Rademacher complexity and thus to classical learning settings that assume independent and identically distributed data, in or-

[1] Departments of Computer Science and Electrical Engineering, Columbia University, New York, NY, USA [2] Massachusetts Institute of Technology, Cambridge, MA, USA [3] OpenAI, San Francisco, CA, USA. Correspondence to: Adam Block <adam.block@columbia.edu>.

*Proceedings of the 43$^{rd}$ International Conference on Machine Learning*, Seoul, South Korea. PMLR 306, 2026. Copyright 2026 by the author(s).

der to allow for dependent and potentially nonstationary data (as well as, more recently, learning under privacy constraints (Golowich, 2021; Golowich & Livni, 2021)), one must instead consider the sequential Rademacher complexity, which is defined by martingale sequences as opposed to sums of independent random variables. In addition, in order to achieve more refined bounds in the case of regression, where exploiting the additional curvature of the squared loss $\ell(x) = x^2$ can accelerate learning, one is led to analyze the offset Rademacher complexity.[1] In both of these cases, tools have been developed to analyze these complexities. Notably, (Rakhlin et al., 2015; Block et al., 2021) use chaining techniques to obtain general comparison estimates for sequential Rademacher complexity. However unlike the classical setting of Gaussian processes, these estimates are not sharp up to constant factors and, in particular, apply only to the *worst-case* complexities and can thus not take advantage of beneficial structure in the data. As a result one pays additional factors depending on the combinatorial complexity of the class $\mathcal{F}$, as well as the number of potentially relevant "scales" in play, leading to loose bounds on the sample complexity. Removing these factors would automatically improve generalization bounds that arise from this theory. In the sequential case, contraction on a per-tree basis can be used to construct algorithms attaining minimax regret as was done in (Rakhlin et al., 2012), while in in the offset case tighter generalization bounds on composed function classes are possible when learning with square loss. Even analogs of equation 1 with additional factors would be useful in cases when these factors are mild. Thus it is natural to ask:

> *Does the Ledoux–Talagrand contraction lemma*
> *extend to sequential and offset Rademacher complexities? If not, how badly can it fail?*

We resolve both cases of the first question affirmatively up to mild additional factors, under symmetry and convexity assumptions on $\mathcal{F}$ summarized below (see Subsection 2 for detailed statements). For the sequential Rademacher complexity, we show that if $\mathcal{F}$ is *symmetric*, then contraction holds up to a factor of $O(d\log^2(n))$, where $d$ is the Littlestone dimension of $\mathcal{F}$, which we show to be qualitatively tight in the sense that contraction must pay at least a factor of $\Omega(d + \log(n))$ even under the symmetry constraint. We further show that removing the symmetry assumption results in an exponentially worse $\Omega(\sqrt{n})$ lower bound on contraction.

In the case of offset Rademacher complexity, we demonstrate that if $\mathcal{F}$ is symmetric and *convex*, then contraction

can be established up to factors polylogarithmic in $n$, and which cannot be removed. Finally, we show that absent convexity, the offset Rademacher complexity can be 0 with the contracted analogue positive, indicating that contraction is not possible in general.

We emphasize that, in applications, most function classes of interest will already be symmetric. For example if $\mathcal{F}$ is a class of deep neural networks, both convexity and symmetry can be enforced in the final linear layer. Even in the case that positivity constraints make $\mathcal{F}$ asymmetric, manually symmetrizing $\mathcal{F}$ to $\mathcal{F} \cup (-\mathcal{F})$ will preserve, up to constants, all covering number bounds that are commonly used to estimate Rademacher complexity. Thus our results indicate that for practical purposes the price of contraction in sequential and regression settings is at worst polylogarithmic in the dataset size $n$.

We begin the paper by introducing the necessary technical prerequisites in Section 1.1 before highlighting a number of related works as well as applications of the sequential and offset Rademacher complexities in Section 1.2. We summarize our main results in Section 2. In the remainder of the paper, we provide general outlines for our results, with those related to sequential Rademacher complexity in Section 3 and those related to offset Rademacher complexity in Section 4.

## 1.1. Preliminaries on Rademacher Complexity

Let $\mathcal{X}$ be any set , $\mathcal{F}$ a class of functions $f : \mathcal{X} \to [-1, 1]$, and $(\epsilon_i)_{i=1}^n \in \{\pm 1\}^n$ a sequence of i.i.d. Rademacher variables. The empirical Rademacher avarages (with respect to $\mathbf{x} = (x_1, \ldots, x_n) \in \mathcal{X}^n$) and the worst case Rademacher complexity $\mathcal{R}_n(\mathcal{F})$ of $\mathcal{F}$ are defined, respectively, as

$$\mathcal{R}_n(\mathcal{F}; \mathbf{x}) = \frac{1}{n}\mathbb{E}_\epsilon \sup_{f \in \mathcal{F}} \sum_{i=1}^n \epsilon_i f(x_i),$$

$$\mathcal{R}_n(\mathcal{F}) = \sup_{\mathbf{x} \in \mathcal{X}^n} \mathcal{R}_n(\mathcal{F}; \mathbf{x}).$$

These quantities are fundamental to generalization bounds. One of its appealing properties is the Ledoux-Talagrand contraction lemma recalled below. It should be noted that the contraction lemma in (Ledoux & Talagrand, 1991) is more general than stated, yet requires $\ell(0) = 0$. The simple proof of Lemma 1.1 first appeared, to the best of our knowledge, in the lecture notes of (Kakade & Tewari, 2008).

**Lemma 1.1.** *Let $\ell : \mathbb{R} \to \mathbb{R}$ be a 1-Lipschitz function. Then for any $\mathbf{x}$,*
$$\mathcal{R}_n(\ell \circ \mathcal{F}; \mathbf{x}) \leq \mathcal{R}_n(\mathcal{F}; \mathbf{x}). \qquad (2)$$

*In particular, maximizing over $\mathbf{x}$ implies $\mathcal{R}_n(\ell \circ \mathcal{F}) \leq \mathcal{R}_n(\mathcal{F})$.*

Since this lemma holds for each fixed $\mathbf{x}$, it also holds on

---

[1]To analyze online learning with squared loss, Rakhlin & Sridharan (2014) introduced the sequential offset Rademacher complexity; our lower bounds apply to this setting as well, but we leave a full analysis of this other notion to future work.

average over e.g. i.i.d. $(x_1, \ldots, x_n) \in \mathcal{X}^n$, which is the relevant setting for generalization on i.i.d. data.

The notion of function class 'size' or complexity can be measured in many different ways, with one of the more popular being **covering numbers**. The covering number of a class $\mathcal{F}$ with respect to the empirical norm $\|\cdot\|_n$ at scale $\epsilon$, denoted by $\mathcal{N}(\mathcal{F}, \|\cdot\|_n, \epsilon)$, is the minimal cardinality of a set $\{f_i\}$ of functions such that every $f \in \mathcal{F}$ is contained in an $\epsilon$-ball around some $f_i$. Dual to covering numbers is the notion of **packing numbers**, where $\mathcal{M}(\mathcal{F}, \|\cdot\|_n, \epsilon)$ is the maximal cardinality of a set of functions in $\mathcal{F}$ such that the pairwise distances within the set are at least $\epsilon$. Typically the statistical rates of learning a function class $\mathcal{F}$ are governed by how quickly the entropy, $\log \mathcal{N}(\mathcal{F}, \|\cdot\|_n, \epsilon)$, grows as $\epsilon \downarrow 0$; parametric function classes typically exhibit growth on the order $\log(1/\epsilon)$, while nonparametric classes can have entropy growth on the order $\epsilon^{-p}$ for some $p > 0$. We say that a class $\mathcal{F}$ is **Donsker** if $\log \mathcal{N}(\mathcal{F}, \|\cdot\|_n, \epsilon) \lesssim \epsilon^{-2}$ for all $\epsilon \in (0,1)$. Donsker function classes are important in the study of empirical processes and asymptotic statistics (Geer, 2000; Van der Vaart, 2000) as they characterize those classes for which a uniform central limit theorem holds.

(Rakhlin et al., 2010) introduced a generalization called **sequential Rademacher complexity**, showing that it controls the adversarial minimax regret in online learning problems. Here $\mathbf{z}$ is a general $\mathcal{X}$-valued stochastic process adapted to $\epsilon$, i.e.

$$\mathbf{z}_i = \mathbf{z}_i(\epsilon_1, \ldots, \epsilon_{i-1}).$$

We will write $\mathbf{z}$ instead of $\mathbf{x}$ from now on to emphasize this distinction. One then similarly defines

$$\mathcal{R}_n^{\mathrm{seq}}(\mathcal{F}; \mathbf{z}) = \frac{1}{n} \mathbb{E}_\epsilon \sup_{f \in \mathcal{F}} \sum_{i=1}^n \epsilon_i f(\mathbf{z}_i), \qquad (3)$$

$$\mathcal{R}_n^{\mathrm{seq}}(\mathcal{F}) = \sup_{\mathbf{z}} \mathcal{R}_n^{\mathrm{seq}}(\mathcal{F}; \mathbf{z}).$$

When $\mathbf{z}$ is a fixed adapted process, $\mathcal{R}_n^{\mathrm{seq}}(\mathcal{F}; \mathbf{z})$ is known as the **tree dependent** sequential Rademacher complexity while $\mathcal{R}_n^{\mathrm{seq}}(\mathcal{F})$ is the **worst-case** sequential Rademacher complexity.

For the worst-case quantity $\mathcal{R}_n^{\mathrm{seq}}(\mathcal{F})$, Lemma 13 in (Rakhlin et al., 2015) shows that

$$\frac{\mathcal{R}_n^{\mathrm{seq}}(\ell \circ \mathcal{F})}{\mathcal{R}_n^{\mathrm{seq}}(\mathcal{F})} \leq O(\log^{3/2}(n)). \qquad (4)$$

In other words, the contraction lemma extends to the sequential setting up to polylogarithmic factors. Additionally, (Block et al., 2021) showed the right-hand side of equation 4 can be taken independent of $n$ if $\mathcal{F}$ obeys natural combinatorial properties. However neither result extends to the tree-dependent quantity $\mathcal{R}_n^{\mathrm{seq}}(\mathcal{F}; \mathbf{z})$, which would give a sequential analog of equation 2. In fact, we will see that both of these extensions are false in general.

We also will require the combinatorial notion of complexity known as **Littlestone dimension** (Littlestone, 1988) and will denote this by $d = \mathrm{Ldim}(\mathcal{F})$ for binary-valued $\mathcal{F}$. The Littlestone dimension, a sequential analogue of the more commonly known VC dimension (Vapnik & Chervonenkis, 1968), is defined to be the maximal $d$ such that there exists a complete, $\mathcal{X}$-labelled binary tree of depth $d$ that is shattered by $\mathcal{F}$, where a tree $\mathbf{z}$ is *shattered* if for all $\epsilon_{1:d} \in \{\pm 1\}^d$, there exists $f \in \mathcal{F}$ such that $f(z_i) = \epsilon_i$ for all $i \in [d]$. More generally, for $\alpha > 0$ we say a $\mathbb{R}$-valued class $\mathcal{F}$ has **sequential scale-sensitive dimension** $\mathrm{sdim}_\alpha(\mathcal{F}) \geq d$ if there exists a $\mathbb{R}$-valued binary tree of depth $d$ such that for all $\vec{\epsilon} \in \{\pm 1\}^d$, there is $f \in \mathcal{F}$ such that for all $1 \leq t \leq d$, we have $\epsilon_t(f(\mathbf{z}_t(\vec{\epsilon}))) \geq \alpha/2$. (Thus $\mathrm{sdim}_\alpha(\mathcal{F})$ is defined to be maximal $d \geq 0$ for which such a tree exists.) This recovers the Littlestone dimension when $\mathcal{F}$ is $\{\pm 1\}$-valued (for any $\alpha \in (0, 1/2)$).

Finally, we consider the **offset Rademacher complexity** with offset parameter $\eta \geq 0$:

$$\mathcal{R}_n^\eta(\mathcal{F}; \mathbf{x}) = \frac{1}{n} \mathbb{E}_\epsilon \sup_{f \in \mathcal{F}} \sum_{i=1}^n [\epsilon_i f(x_i) - \eta f(x_i)^2], \qquad (5)$$

We write $\mathcal{R}_n^\eta(\mathcal{F})$ when the dataset $\mathbf{x}$ is clear from context. This notion was used (with the centered class $\mathcal{F} - f^*$) in (Liang et al., 2015) to study statistical learning under squared loss, where the offset allows the complexity to 'localize' around $f^*$ and allows for faster rates of estimation than an analysis relying solely on the standard Rademacher complexity. Intuitively, the squared loss penalizes those $f \in \mathcal{F}$ with large $L^2$ norm, leading to better upper bounds for more positive $\eta > 0$. Note that $\eta = 0$ recovers the usual Rademacher complexity.

## 1.2. Related Work

**Sequential Rademacher Complexity.** Sequential versions of the Rademacher complexity were developed in (Rakhlin et al., 2010; 2011; 2015). Such quantities bound the minimax value of very general online learning problems with adaptive adversaries, often leading to optimal algorithms (Rakhlin et al., 2012). These complexities and their variants have been used to study a wide range of topics including reinforcement learning (Dong et al., 2021; Jin et al., 2022), non-stationary time-series (Kuznetsov & Mohri, 2015; 2017), improper logistic regression (Foster et al., 2018), causal inference (Gao et al., 2022), characterizing online learnability (Alon et al., 2021), online regression in Hilbert spaces (Subedi et al., 2024), martingale tail bounds in Banach space (Rakhlin & Sridharan, 2017), online top-$k$ optimization (Bubeck et al., 2021), smoothed online learning (Block et al., 2022; Block & Polyanskiy, 2023), and differentially private learning (Golowich, 2021; Golowich & Livni, 2021). There has also been further math-

ematical development of the theory, including the $n \to \infty$ asymptotics for fixed finite $\mathcal{F}$ (Rokhlin, 2017) and the behavior of such quantities in multi-class settings (Hanneke et al., 2023). Contraction inequalities for *worst-case* (i.e., over possible trees) sequential Rademacher complexity were established with a polylogarithmic factor in (Rakhlin et al., 2010) and more tightly but under stricter complexity assumptions in (Block et al., 2021). To our knowledge, the question of contraction for *tree-dependent* sequential Rademacher complexity has not been previously considered.

**Offset Rademacher Complexity.** Offset Rademacher complexity was introduced in (Liang et al., 2015), and it can be shown to be related to the "localized" versions of Rademacher complexity (Koltchinskii & Panchenko, 2000; Bousquet et al., 2002; Bartlett et al., 2005). The localized quantities led to more refined generalization bounds by only considering the fluctuations of the empirical (or Rademacher) process on a relevant subset of the class $\mathcal{F}$. While (Liang et al., 2015) considered only the squared loss, (Vijaykumar, 2021) later extended the ideas to a general family of so-called $(\mu, d)$-convex losses, which encompass uniform convexity, exponential concavity, self-concordance, and $p$-th powers. Offset Rademacher complexity has been applied to the analysis of early-stopped mirror descent (Vaskevicius et al., 2020), time-series models (Ziemann & Tu, 2022), empirical Bayes estimation (Jana et al., 2023), and deep learning (Mou et al., 2018; Duan et al., 2023). Sequential analogues have also been introduced (Rakhlin & Sridharan, 2014) and the theory has been further developed in several directions by e.g. (Foster et al., 2015; Zhivotovskiy & Hanneke, 2018; Block et al., 2024). More recently, a relaxation of offset Rademacher complexity, known as the *Will's Functional*, has been studied in (Mourtada, 2023); this complexity is known to satisfy contraction with neither additional assumptions on $\mathcal{F}$ nor additional constant or polylogarithmic factors. In this work, we restrict our focus to the original offset Rademacher complexity as opposed to its sequential analogue; note that our lower bounds in Theorems 2.5 and 2.6 hold even for the tree-dependent case, but we leave the question of upper bounds in this setting to future work. We also note that a variant of the contraction lemma for offset Rademacher complexity where the Lipschitz function $\ell$ is applied only to the first term in the sum was established as Mou et al. (2018, Lemma 1); while useful for their applications this variant of $\mathcal{R}_n^\eta(\mathcal{F})$ is not as natural due to the $\mathcal{F}$-dependent nature of the offset; furthermore, such a contraction inequality cannot be used to establish further structural properties of the complexity, such as monotonicity, which require $\ell$ to be applied to both terms.

## 2. Main Results

We study the extent to which the contraction inequality can be generalized beyond classical Rademacher complexity. We begin by considering the sequential case and then proceed to the offset setting.

### 2.1. Sequential Rademacher Complexity

Unfortunately, the short and elegant proof of equation 1 found in (Kakade & Tewari, 2008) cannot be extended to sequential Rademacher complexity because of the complex dependence among the $\epsilon_j$ and $\mathbf{z}_i$. Furthermore, the approaches in (Rakhlin et al., 2015; Block et al., 2021) relate $\mathcal{R}^{\mathrm{seq}}(\mathcal{F})$ to worst-case combinatorial quantities that do satisfy contraction and thus cannot be extended to the tree-dependent complexity under consideration here. We now provide the first positive result showing that contraction of $\mathcal{R}_n^{\mathrm{seq}}(\mathcal{F}, \mathbf{z})$ can be established up to factors polynomial in the sequential scale-sensitive dimension and polylogarithmic in $n$ whenever $\mathcal{F}$ is *symmetric*. We say the class $\mathcal{F}$ is *C-bounded* if it is $[-C, C]$-valued, and *C-integral* if it is $[-C, C] \cap \mathbb{Z}$ valued.

**Theorem 2.1.** *Let $\mathcal{F}$ be $C$-integral and symmetric with $\mathrm{sdim}_\alpha(\mathcal{F}) = d$ for some $\alpha \in (0, 1/2)$. Suppose $\ell : \mathbb{R} \to \mathbb{R}$ satisfies $|\ell(x)| \le |x|$ for all $x \in \mathbb{R}$. Then,*

$$\mathcal{R}_n^{\mathrm{seq}}(\ell \circ \mathcal{F}, \mathbf{z}) \le O\big(C^2 d^{3/2} \log^{5/2}(Cn)\big) \mathcal{R}_n^{\mathrm{seq}}(\mathcal{F}, \mathbf{z}) \quad (6)$$

In Appendix A, we generalize the above result to general, $C$-bounded function classes through discretization in Theorem A.1, but defer the statement for the sake of space. We emphasize that Theorem 2.1 applies more generally than just Lipschitz functions: indeed, as all versions of Rademacher complexity are invariant under constant additions due to the fact that Rademacher random variables are are mean zero, given a lipschitz function $\ell$, we can form $\ell'(x) = \ell(x) - \ell(0)$, which satisfies $|\ell'(x)| \le |x|$ without changing the left hand side of equation 6. As mentioned previously, Theorem 2.1 can be used to construct algorithms with small regret through the relaxations approach of (Rakhlin et al., 2012); for example, if $\mathbf{z}$ is a tree with monotone paths (i.e. $\mathbf{z}_i \ge \mathbf{z}_j$ for all $i \ge j$) and $\mathcal{F}$ is the class of one dimensional thresholds, then worst-case sequential Rademacher complexity does not decay (Littlestone, 1988) but equation 6 can still lead to a nontrivial bound.

We now show that Theorem 2.1 is qualitatively tight, both in its assumptions and the factors in the bound. First, we show that the symmetry assumption is necessary in order to obtain a nontrivial contraction factor.

**Theorem 2.2.** *There exists a class $\mathcal{F}$ of $\{0, 1\}$-valued functions with Littlestone dimension $d = 1$ and tree $\mathbf{z} \in \mathcal{Z}^n$*

*such that with $\ell(x) = -x$:*

$$\sup_{\epsilon \in \{\pm 1\}^n} \sup_{f \in \mathcal{F}} \sum_{i=1}^{n} \epsilon_i f(z_i) \leq 1, \tag{7}$$

$$\mathbb{E}_\epsilon \sup_{f \in \mathcal{F}} \sum_{i=1}^{n} \epsilon_i \ell(f(z_i)) \geq \Omega(\sqrt{n}).$$

Note that the first estimate in equation 7 is even stronger than necessary, as the upper bound on the martingale process holds *almost surely* in the Rademacher random variables as opposed to merely in expectation. Theorem 2.2 shows that, absent symmetry, contraction can be exponentially worse (scaling polynomially in $n$ as opposed to polylogarithmically) than in the setting of Theorem 2.1. Finally, we show that in addition to the symmetry assumption, neither the Littlestone dimension nor polylogarithmic factor in $n$ found in equation 6 can be removed, even when $\mathcal{F}$ is symmetric.

**Theorem 2.3.** *There exists a symmetric function class $\mathcal{F}$ : $\mathcal{X} \to \mathbb{R}$ with Littlestone dimension $d = 1$ and a tree $\mathbf{z} \in \mathcal{Z}^n$ such that there is some Lipschitz contraction $\ell : \mathbb{R} \to \mathbb{R}$ such that:*

$$\sup_{\epsilon \in \{\pm 1\}^n} \sup_{f \in \mathcal{F}} \sum_{i=1}^{n} \epsilon_i f(z_i) \leq O(1), \tag{8}$$

$$\mathbb{E}_\epsilon \sup_{f \in \mathcal{F}} \sum_{i=1}^{n} \epsilon_i \ell(f(z_i)) \geq \Omega(\log n).$$

*Furthermore, for any $1 \leq d \leq n$, there exists a symmetric function class $\mathcal{F}$ with $\mathrm{Ldim}(\mathcal{F}) = d$, tree $\mathbf{z} \in \mathcal{Z}^n$, and Lipschitz contraction $\ell$ such that*

$$\sup_{\epsilon \in \{\pm 1\}^n} \sup_{f \in \mathcal{F}} \sum_{i=1}^{n} \epsilon_i f(z_i) \leq O(1), \qquad and$$

$$\inf_{\epsilon \in \{\pm 1\}^n} \sup_{f \in \mathcal{F}} \sum_{i=1}^{n} \epsilon_i \ell(f(z_i)) \geq \Omega(d).$$

In the latter case, we emphasize that both the upper bound on $\mathcal{R}_n^{\mathrm{seq}}(\mathcal{F}, \mathbf{z})$ and the lower bound on $\mathcal{R}_n^{\mathrm{seq}}(\ell \circ \mathcal{F}, \mathbf{z})$ hold *almost surely* in the Rademacher random variables.

## 2.2. Offset Rademacher Complexity

Having established a qualitatively tight understanding of contraction of tree-dependent sequential Rademacher complexity, we now turn to the case of offset Rademacher complexity. In this case, we require both symmetry and *convexity* of $\mathcal{F}$ in order to obtain a result, as well as a technical assumption on the complexity of the class.

**Theorem 2.4.** *Let $\mathcal{F}$ : $\mathcal{X} \to \mathbb{R}$ be a convex, symmetric, Donsker function class and suppose that $\eta \leq (4\sqrt{\log(n)})^{-1}$. Then it holds that for any contraction $\phi$,*

$$\mathcal{R}_n^\eta(\phi \circ \mathcal{F}) \lesssim \sqrt{\log(n)} \cdot (1 + 1/\eta) \cdot \mathcal{R}_n^\eta(\mathcal{F}).$$

Due to the restriction on $\eta$, the best possible upper bound is $\mathcal{R}_n^\eta(\phi \circ \mathcal{F}) \lesssim \log(n) \cdot \mathcal{R}_n^\eta(\mathcal{F})$. For handling larger $\eta$, we can always bound $\mathcal{R}_n^\eta(\cdot) \leq \mathcal{R}_n^{\eta'}(\cdot)$ for $\eta' \leq \eta$ and then apply the above theorem for sufficiently small $\eta'$ to get an upper bound of $\sqrt{\log(n)}/\eta \cdot \mathcal{R}_n^{\eta'}(\mathcal{F})$. We emphasize that this still suffices to obtain complexities that are polylogarithmic in the number of samples, corresponding to the 'fast rates' of regression, unlike in the case of standard Rademacher complexity which must be at least polynomial in $n$ and thus this assumption does not trivialize the result. Indeed, in chaining upper bounds for the offset Rademacher complexity (cf. eg. Liang et al. (2015, Lemma 6)), the upper bound scales as $\eta^{-1}$ and thus we pay at most a polylogarithmic factor in $n$ for this assumption; as such a factor already appears in the contraction result, this is not a significant restriction. One setting where Theorem 2.5 can be applied is the case of generalized linear models where $\mathcal{F} = \{x \mapsto \phi(\langle w, x \rangle) | w \in \mathbb{R}^d \text{ and } \|w\| \leq B\}$ for some Lipschitz function $\phi$; as the offset Rademacher complexity of the class of linear functions is well-understood due to (Liang et al., 2015), Theorem 2.4 immediately yields statistical rates of regression for $\mathcal{F}$.

As in the case of offset Rademacher complexity, we demonstrate the necessity of several assumptions appearing in Theorem 2.4. First, we show that absent symmetry and convexity, no contraction inequality is possible.

**Theorem 2.5.** *For any $\eta > 0$ and any $n = \Omega(1/\eta)$, there is a uniformly bounded function class $\mathcal{F}$ and Lipschitz function $\ell$ such that $\mathcal{R}_n^\eta(\mathcal{F}) = 0$ but $\mathcal{R}_n^\eta(\ell \circ \mathcal{F}) > 0$. In particular, no contraction inequality can hold, even if the constant is allowed to depend arbitrarily on $n$.*

The construction in Theorem 2.5 critically requires $n = \Omega(1/\eta)$, which is easily seen to be necessary. Indeed, applying the contraction inequality of the standard Rademacher complexity, equation 1, we see that $\mathcal{R}_n^\eta(\ell \circ \mathcal{F}) \leq \mathcal{R}_n^\eta(\mathcal{F}) + O(\eta)$, assuming uniform boundedness. Thus, if $\eta \lesssim \mathcal{R}_n^\eta(\mathcal{F})$ then contraction holds trivially. As $\mathcal{R}_n^\eta(\mathcal{F})$ is typically $\Theta(n^{-1})$ for parametric classes (Liang et al., 2015), we require $n = \Omega(1/\eta)$ to ensure such a lower bound. Understanding the extent to which contraction is possible absent the Donsker assumption is an interesting question we leave to future work.

Finally, we demonstrate that Theorem 2.4 is qualitatively tight, in that no contraction inequality with constant better than $\Omega(\log n)$ is possible.

**Theorem 2.6.** *For any fixed $\eta > 0$ and $n$ large there exists a convex, symmetric class $\mathcal{F}$ of functions $\mathcal{X} \to [-1, 1]$ and 1-Lipschitz $\ell : \mathbb{R} \to \mathbb{R}$ such that*

$$\mathcal{R}_n^\eta(\ell \circ \mathcal{F}; \mathbf{x}) \geq \Omega(\log n) \cdot \mathcal{R}_n^\eta(\mathcal{F}; \mathbf{x}). \tag{9}$$

*Further, one can choose a probability distribution over $\mathcal{X}$*

*such that for $x_1, \ldots, x_n$ IID from this distribution,*

$$\mathbb{E}\mathcal{R}_n^\eta(\ell \circ \mathcal{F}; \mathbf{x}) \geq \Omega(\log n) \cdot \mathbb{E}\mathcal{R}_n^\eta(\mathcal{F}; \mathbf{x}). \quad (10)$$

*Finally in both of the above results, it suffices to use the 1-dimensional class*

$$\mathcal{F} = \{tf_* \ : \ t \in [-1,1]\} \quad (11)$$

*for some fixed $f_* : \mathcal{X} \to [0.5, 1]$ (not depending on $\mathbf{x}$).*

The construction here is of a very different flavor than the results in the sequential case, as we describe in the sequel. This result also implies that, as stated, Theorem 2.4 cannot be extended to constant $\eta$, at least without removing the square root on the logarithmic factor. Whether or not such an extension is possible with a different polynomial in $\log(n)$ remains an interesting open question.

# 3. Proofs for Sequential Rademacher Complexity

In this section, we prove Theorems 2.1 and 2.2 and defer the proof of Theorem 2.3 to Appendix C for the sake of space. Our upper bound relies on identifying a different measure of the size of $\mathcal{F}$ for which contraction is easy to prove and then demonstrating that this measure approximates sequential Rademacher complexity in the case of symmetric $\mathcal{F}$. Our lower bounds rely on careful analyses of random walks on the integers, with the $\sqrt{n}$ arising from the fact that a random walk started at zero is $\Theta(\sqrt{n})$ in magnitude after $n$ steps.

We introduce the following notation: for $s \leq n$, let

$$\mathbb{T}_{\mathbf{v},s}(\epsilon) = \mathbb{T}_{\mathbf{v}}(\epsilon_1, \ldots, \epsilon_s) = \sum_{t=1}^{s} \epsilon_t \mathbf{v}_t(\epsilon_1, \ldots, \epsilon_{t-1})$$

denote the *tree process* associated to signs $\epsilon_{1:s} \in \{\pm 1\}^s$ and tree $\mathbf{v}$. When $\mathbf{z}$ is fixed, we write

$$\mathbb{T}_{f,s} = \mathbb{T}_{f(\mathbf{z}),s} = \sum_{t=1}^{s} \epsilon_t f(\mathbf{z}_t(\epsilon_1, \ldots, \epsilon_{t-1})). \quad (12)$$

## 3.1. Proof Sketch of Theorem 2.1

We defer a full proof of Theorem 2.1 to Appendix A, along with a more general statement, due to space constraints, but we sketch the main idea here. For the sake of simplicity, we consider the setting of $|\mathcal{F}| < \infty$ here as this illustrates the main idea; the more general setting can be found in the rigorous proof. The main idea of the proof is to relate the sequential Rademacher complexity to another measure of size for which demonstrating contraction is easy. Motivated by the classic Burkholder-Davis-Gundy inequalities (Burkholder et al., 1972), we consider the (predictable)

*quadratic variation*, defined to be

$$\langle \mathbb{T}_f \rangle_n = \sum_{t=1}^{n} f(\mathbf{z}_t(\epsilon_1, \ldots, \epsilon_{t-1}))^2.$$

It is easy to see that for each $f$, contraction of the quadratic variation holds, i.e., $\langle \mathbb{T}_{\ell \circ f} \rangle_n \leq \langle \mathbb{T}_f \rangle_n$ for all $n$; indeed, this holds beyond the case of Lipschitz $\ell$ and is why we can prove a more general contraction inequality. The challenge is to relate $\sup_{f \in \mathcal{F}} \langle \mathbb{T}_f \rangle_n$ to $\mathcal{R}_n^{\mathrm{seq}}(\mathcal{F}, \mathbf{z})$, after which the result will follow. The first step is to prove the following result, showing that the expected supremum of the tree process is approximately on the same order as the running maximum.

**Lemma 3.1.** *Suppose that $\mathcal{F}$ is $C$-integral and let $\mathbf{z}$ be a fixed binary tree of depth $n$. Then with $\lesssim$ denoting constant multiplicative factors,*

$$\frac{\mathbb{E}\left[\sup_{f \in \mathcal{F}} \max_{t \leq n} |\mathbb{T}_{f,t}|\right]}{\log(Cn)} \lesssim \mathbb{E}\left[\sup_{f \in \mathcal{F}} |\mathbb{T}_{f,n}|\right],$$

$$\mathbb{E}\left[\sup_{f \in \mathcal{F}} |\mathbb{T}_{f,n}|\right] \leq \mathbb{E}\left[\sup_{f \in \mathcal{F}} \max_{t \leq n} |\mathbb{T}_{f,t}|\right]$$

The right hand side of the above is immediate, and the left hand side follows from a careful analysis of the tail behavior of the tree process. The second step is to relate the quadratic variation to the running maximum of the tree process, which we do in the following lemma.

**Lemma 3.2.** *Suppose that $\mathcal{F}$ is $C$-integral and $\mathbf{z}$ is a binary tree of depth $n$. Then it holds with probability at least $1 - \delta$ that*

$$\frac{\langle \mathbb{T}_f \rangle_n}{\log(C|\mathcal{F}|/\delta)} \lesssim \sup_{f \in \mathcal{F}} \max_{t \leq n} |\mathbb{T}_{f,t}| \lesssim \log(C|\mathcal{F}|/\delta) \langle \mathbb{T}_f \rangle_n.$$

This result, proved in several steps in Appendix A, relies on self-normalized martingale inequalities and the tail behavior of escape times of martingales from bounded strips of the real line. Note that Lemmas 3.1 and 3.2 involve $|\mathbb{T}_{f,n}|$; it is for this reason that the symmetry of $\mathcal{F}$ is essential, as $\sup_{f \in \mathcal{F}} |\mathbb{T}_{f,n}| = \sup_{f \in \mathcal{F}} \mathbb{T}_{f,n}$ holds almost surely when $\mathcal{F}$ is symmetric, with the expectation of the latter being the sequential Rademacher complexity. The result concludes by combining these two lemmas with the contraction of the quadratic variation to obtain the desired result. The general case of infinite and real-valued $\mathcal{F}$ can be handled by discretization, as we show in Appendix A. It is here that the dimension of $\mathcal{F}$ enters, to bound the effective size of $\mathcal{F}$ restricted to relevant inputs $\mathbf{z}_t$.

## 3.2. Proof of Theorem 2.2

To prove Theorem 2.2, we consider the simple function class $\mathcal{F} = \{f_1, f_2, \dots\}$ given by:

$$Z = \mathbb{Z}_{\geq 1}, \quad f_i(z) = 1_{z=i}, \quad \forall\, (i, z) \in \mathbb{Z}_{\geq 0} \times \mathbb{Z}_{\geq 1}. \tag{13}$$

In particular note that $f_0(z) \equiv 0$ is the zero function. It is easy to see that $\mathcal{F}$ thus defined satisfies $d = 1$, because if $f, f' \in \mathcal{F}$ satisfy $f(z) > f'(z)$ then $f = f_z$.

Theorem 2.2 follows from the following intuitive lemma, proved in Appendix B:

**Lemma 3.3.** *Let $0 = a_0, a_1, \dots, a_n$ be a simple random walk on $\mathbb{Z}$ with sticky reflecting barrier at $0$, given by $a_{i+1} = \max(a_i + \sigma_{i+1}, 0)$ for an i.i.d. sequence of Rademacher signs $\sigma_1, \dots, \sigma_n \in \{\pm 1\}$. Then $\mathbb{E}[a_n] = \Theta(\sqrt{n})$.*

*Proof of Theorem 2.2.* Let $\mathbf{z}_0 = 1$ and define $\mathbf{z}_t(\epsilon_1, \dots, \epsilon_{t-1})$ as follows. If $\mathbb{T}_{f_{\mathbf{z}_{t-1}}, t-1} = 1$, then set $\mathbf{z}_t = \mathbf{z}_{t-1} + 1$; otherwise, set $\mathbf{z}_t = \mathbf{z}_{t-1}$. Note that $\mathbb{T}_{f_{\mathbf{z}_0}, 0} = \mathbb{T}_{f_1, 0} = 0 \neq 1$, hence $\mathbf{z}_1 = \mathbf{z}_0 = 1$. By the definition of $\mathcal{F}$ in equation 13, for each $i$, the stochastic process $\mathbb{T}_{f_i, t}$ changes only while $\mathbf{z}_t = i$. The update rule for $\mathbf{z}_t$ thus means that first $\mathbb{T}_{f_1, t}$ changes until reaching $1$ (since $\mathbf{z}_t = 1$), then $\mathbb{T}_{f_2, t}$ starts changing until reaching $1$ (since $\mathbf{z}_t = 2$), and so on. Thus the first part of equation 7 holds by construction since in fact $\sup_{i,t} \mathbb{T}_{f_i, t} \leq 1$. On the other hand, it is immediate that the random process $t \mapsto \mathbb{T}_{\ell \circ f_{\mathbf{z}_{t-1}}, t-1} = \mathbb{T}_{-f_{\mathbf{z}_{t-1}}, t-1}$ evolves according to the process in Lemma 3.3, due to the changing values of $\mathbf{z}_t$. Thus the second part of equation 7 holds as well, concluding the proof. $\square$

The intuition behind the above construction is that the $\mathbb{T}_{\ell \circ f, t-1}$ will have smaller absolute increments than $\mathbb{T}_{f, t-1}$ for any $f \in \mathcal{F}$ whenever $|\ell(x)| \leq x$ and thus crucially relies on the fact that $\mathbb{T}_{\ell \circ f_i, t-1}$ nonetheless has a good chance to be larger than $\mathbb{T}_{f_i, t-1}$ by virtue of the associated random walk staying positive for all time; thus, a similar idea will not hold for symmetric $\mathcal{F}$, explaining the lack of contradiction with Theorem 2.1.

While we defer a proof of Theorem 2.3 to Appendix C, we note that it follows from a related construction involving random walks on the integers, but forces us to consider only symmetric $\mathcal{F}$. To yield the $\log(n)$ lower bound, we use the fact that after $n$ steps of a random walk with increments distributed as random signs, there is a string of length $\Theta(\log(n))$ of consecutive same signs with constant probability; the Littlestone dimension lower bound follows from a similar, but more involved construction.

# 4. Proof Ideas for Offset Rademacher Complexity

In this section, we explain some of the ideas in our proofs related to the offset Rademacher complexity. For the sake of space, we defer several technical results to the appendix. We begin with our upper bound, which follows from relating the offset Rademacher complexity to the minimax rates of regression with square loss, up to logarithmic factors (Liang et al., 2015) We then continue by providing our construction for the lower bound against any contraction holding in the general case for large $n$. Due to the technical nature of the analysis, we defer a discussion and proof of Theorem 2.6, which relies on a subtle construction of the function class $\mathcal{F}$ and an application of the Berry–Esseen approximation to the central limit theorem, to Appendix E.

### 4.1. Proof Sketch of Theorem 2.4

The intuition for the proof of the upper bound is similar to that in the sequential case in the sense that we identify a different notion of complexity for which it is easy to show contraction holds, and then demonstrate that this alternative notion is approximately equivalent to offset Rademacher complexity. Here, we use a result of (Liang et al., 2015) to relate the offset Rademacher complexity to the minimax rates of regression with square loss, up to logarithmic factors and demonstrate that these minimax rates satisfy the desired contraction. The key quantity in this analysis is the *critical radius* of $\mathcal{F}$, defined below.

**Definition 4.1.** Given a function class $\mathcal{F} : \mathcal{X} \to \mathbb{R}$ and a data set $x_1, \dots, x_n$, we define a critical radius as follows. Let $r \mapsto \mathcal{N}(r)$ denote a non-increasing, right continuous function such that $\log \mathcal{N}(\mathcal{F}, \|\cdot\|_n, r) \leq \mathcal{N}(r)$ for all $r > 0$. We then let

$$\tilde{r}_n(\mathcal{F}) = \sup \left\{ r \geq 0 \,|\, nr^2 + 2\log(2) \leq \mathcal{N}(r) \right\}.$$

The main thrust of the proof will be to show that for an appropriate upper bound $\mathcal{N}$ on the log covering number of $\mathcal{F}$, there is a critical radius $\tilde{r}_n(\mathcal{F})$ such that for $\eta \lesssim \log^{-1/2}(n)$, it holds that

$$\frac{\tilde{r}_n(\mathcal{F})^2}{\text{polylog}(n)} \lesssim \mathcal{R}_n^\eta(\mathcal{F}) \lesssim \tilde{r}_n(\mathcal{F})^2. \tag{14}$$

The easy fact that $\tilde{r}_n$ satisfies the desired contraction property is contained in the following lemma, whose proof can be found in Appendix D.

**Lemma 4.2.** *Let $\mathcal{F} : \mathcal{X} \to \mathbb{R}$ denote a function class and suppose that $x_1, \dots, x_n \in \mathcal{X}$ is a dataset. Let $\tilde{r}_n(\mathcal{F})$ denote the critical radius of $\mathcal{F}$ as in Definition 4.1 and suppose that $\phi : \mathbb{R} \to \mathbb{R}$ is 1-Lipschitz. Then it holds that $\tilde{r}_n(\phi \circ \mathcal{F}) \leq \tilde{r}_n(\mathcal{F})$.*

In order to establish equation 14, we need to prove both inequalities. The first is the content of the following lemma, whose formal statement and proof are deferred to Appendix D and is a consequence of a result of Liang et al. (2015).

**Lemma 4.3** (Informal statement of Lemma D.2). *Suppose that $\mathcal{F} : \mathcal{X} \to \mathbb{R}$ is Donsker. Then,*

$$\mathcal{R}_n^\eta(\mathcal{F}) \leq \left(\frac{2}{\eta} + 24\right) n\tilde{r}_n(\mathcal{F})^2.$$

Note that it is in this lemma that the assumption that $\mathcal{F}$ is Donsker is necessary for our proof to go through. As discussed below, Lemma 4.3 cannot be extended beyond Donsker classes. The more challenging direction is the lower bound, where the convexity of $\mathcal{F}$ is required.

**Lemma 4.4** (Informal statement of Lemma D.3). *Suppose that $\mathcal{F} : \mathcal{X} \to [-1, 1]$ is convex and $\eta \lesssim \log^{-1/2}(n)$. Then,*

$$\mathcal{R}_n^\eta(\mathcal{F}) \geq \frac{n\tilde{r}_n(\mathcal{F})^2}{2^{18} \cdot C^4 \sqrt{\log(n)}}.$$

Lemma 4.4 follows by lower bounding $\mathcal{R}_n^\eta(\mathcal{F})$ by the statistical rate of empirical risk minimization in square loss. It is here that the convexity of $\mathcal{F}$ is critical, as the offset Rademacher complexity controls the performance of the more general *star algorithm* studied in (Liang et al., 2015), which reduces to ERM only in the presence of convexity. The proof of Theorem 2.4 then follows by combining Lemma 4.3 and Lemma 4.4 with the contraction property of $\tilde{r}_n$ given in Lemma 4.2. We remark that the proof method of Lemma 4.4 demonstrates that the Donsker assumption in Lemma 4.3 cannot be removed as it is known that the minimax rate of regression with square loss, controlled by $\tilde{r}_n(\mathcal{F})^2$, is not achievable by ERM for non-Donsker classes (Birgé & Massart, 1993). Thus, any approach to proving contraction for more complex function classes will likely require a different approach than the one taken here. The complete proof is deferred to Appendix D.

### 4.2. Proof of Theorem 2.5

We begin by presenting our lower bound construction for general function classes. Let $\mathcal{F} : \mathcal{X} \to \mathbb{R}$ be an arbitrary function class uniformly bounded in absolute value by 1 such that $\mathcal{R}_k^\eta(\mathcal{F}) > 0$ for all $k > 0$; an example of such can be found in linear classes by Lemma 10 in (Liang et al., 2015). Let $\tilde{\mathcal{X}} = \mathcal{X} \cup \{x^\star\}$ and extend $\mathcal{F}$ such that $f(x^\star) = 2$ for all $f \in \mathcal{F}$. Let $m \geq 0$ and let $n = m + k$ such that $x_{k+1} = \cdots = x_{k+m} = x^\star$. Then,

$$n \cdot \mathcal{R}_n^\eta(\mathcal{F}) = \mathbb{E}_\epsilon \left[\sup_{f \in \mathcal{F}} \sum_{i=1}^n \epsilon_i f(x_i) - \eta f(x_i)^2\right]$$

$$= \mathbb{E}_\epsilon \left[\sup_{f \in \mathcal{F}} \sum_{i=1}^k \epsilon_i f(x_i) - \eta f(x_i)^2\right] - 4\eta m$$

$$= k \cdot \mathcal{R}_k^\eta(\mathcal{F}) - 4\eta m.$$

Let $\ell$ be an arbitrary Lipschitz function such that $\ell(t) = t$ for $t \in [-1, 1]$ and $\ell(2) = 0$; an example of such a function takes $\ell(t) = 1 - t$ for $t \in [1, 2]$. Then we have

$$n \cdot \mathcal{R}_n^\eta(\ell \circ \mathcal{F}) = k \cdot \mathcal{R}_k^\eta(\mathcal{F}).$$

If $m = k \cdot \mathcal{R}_k^\eta(\mathcal{F})/\eta$, then $\mathcal{R}_n^\eta(\mathcal{F}) = 0$ but $\mathcal{R}_n^\eta(\ell \circ \mathcal{F}) = \frac{k}{n}\mathcal{R}_n^\eta(\mathcal{F}) = \frac{\eta \mathcal{R}_k^\eta(\mathcal{F})}{\eta + \mathcal{R}_k^\eta(\mathcal{F})} > 0$. Thus, the result follows by setting $n = k + k\mathcal{R}_k^\eta(\mathcal{F})/\eta$. $\square$

As noted above, the construction in the lower bound requires that $n$ grows with $k \cdot \mathcal{R}_k^\eta(\mathcal{F})/\eta$, which can be taken to be $\Theta(1/\eta)$ by assuming $k$ is constant. While this lower bound is unfortunate, it is perhaps unsurprising given the fact that the offset Rademacher process, under whose expectation and supremum $\mathcal{R}_n^\eta(\mathcal{F})$ is defined, is *monotone decreasing* in $n$ for any fixed $f$. We now proceed to sketch the proof of our upper bound for convex function classes.

## 5. Discussion

In this paper, we provided general conditions for which approximate extensions of the Ledoux-Talagrand contraction lemma hold and demonstrate that many of these conditions are necessary. Furthermore, we prove that the factors incurred with contraction are qualitatively tight even under these general conditions. Our results leave open the question of extending contraction inequalities to distribution-dependent sequential Rademacher complexity (Rakhlin et al., 2011), sequential offset Rademacher complexity (Rakhlin & Sridharan, 2014), and quantitatively tight bounds on the extent to which contraction fails. Furthermore, an interesting future direction would be to better understand the behavior of contractions on the worst-case Rademacher complexities as in equation 3. While (Block et al., 2021) demonstrate that such contraction holds under technical assumptions on the function class, the general case remains open.

We note that the fact that contraction does not hold for offset Rademacher complexity suggests that the Will's functional of (Mourtada, 2023), which satisfies contraction along with other nice structural properties and precisely characterizes the rates of square loss regression under fixed design, is more amenable to analysis and control.

## Impact Statement

This paper presents work whose goal is to advance the field of Machine Learning. There are many potential societal consequences of our work, none which we feel must be specifically highlighted here.

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

# A. Proof of Theorem 2.1

In this section, we prove a more general version of Theorem 2.1, Theorem A.1, which extends beyond the $C$-integral special case. The result in the body is the first statement in the following result.

**Theorem A.1.** *Let $\mathcal{F}$ be $C$-bounded and symmetric with $\mathrm{sdim}_\alpha(\mathcal{F}) = d$ for some $\alpha \in (0, 1/2)$. Suppose $\ell : \mathbb{R} \to \mathbb{R}$ satisfies $|\ell(x)| \leq |x|$ for all $x \in \mathbb{R}$.*

1. *If $\mathcal{F}$ is $C$-integral, then*
$$\mathcal{R}_n^{\mathrm{seq}}(\ell \circ \mathcal{F}, \mathbf{z}) \leq O\big(C^2 d^{3/2} \log^{5/2}(Cn)\big) \mathcal{R}_n^{\mathrm{seq}}(\mathcal{F}, \mathbf{z}).$$

2. *If $|\ell(x) - \ell(y)| \leq \delta$ for all $x, y \in [-C, C]$ with $|x - y| \leq \alpha$, then we have the estimates*
$$\mathcal{R}_n^{\mathrm{seq}}(\ell \circ \mathcal{F}, \mathbf{z}) \leq O\left((C/\alpha)^2 d^{3/2} \log^{5/2}(Cn/\alpha)\right) \cdot (\mathcal{R}_n^{\mathrm{seq}}(\mathcal{F}, \mathbf{z}) + \alpha) + \delta;$$
$$\mathcal{R}_n^{\mathrm{seq}}(\ell \circ \mathcal{F}, \mathbf{z}) \leq O\left(d^{3/2} \log^{5/2}(Cn/\alpha)\right)(\mathcal{R}_n^{\mathrm{seq}}(\mathcal{F}, \mathbf{z}) + \alpha) + O\left(\frac{Cd \log(Cn/\alpha)}{n}\right) + \delta. \tag{15}$$

The proof of Theorem A.1 relies on two key intermediate facts. The first is to use sequential analogues of covering numbers from (Rakhlin et al., 2015) to reduce the problem to consider only a finite class of functions. The second then uses the fact that for 'reasonable' discrete-time martingales, the quadratic variation is a good proxy for the running maximum of the absolute value. Informally, we have that with high probability:
$$\sup_{t \leq n} |M_t| \asymp \langle M \rangle_n^{1/2}, \tag{16}$$

where
$$\langle M \rangle_n = \sum_{t \leq n} \mathbb{E}^{t-1}[(M_t - M_{t-1})^2]$$

is the predictable quadratic variation of the martingale $M$. The reason equation 16 is helpful to us is the simple estimate
$$\langle \mathbb{T}_{\ell \circ f} \rangle_n \leq \langle \mathbb{T}_f \rangle_n, \tag{17}$$

where
$$\langle \mathbb{T}_f \rangle_n = \sum_{t=1}^{n} f(\mathbf{z}_t(\epsilon_1, \ldots, \epsilon_{t-1}))^2$$

denotes the predictable quadratic variation of the martingale $\mathbb{T}_f$ from equation 12. Indeed equation 17 follows from the condition $|\ell(x)| \leq |x|$, due to the symmetric nature of the increments in $\mathbb{T}_f$. We note that using $\langle M \rangle_n^{1/2}$ as an approximation for $\sup_{t \leq n} |M_t|$ is also the theme of the famous Burkholder–Davis–Gundy inequalities (see e.g. (Burkholder et al., 1972)), which inspired our approach.

Below we say a discrete-time martingale $M$ is $C$-*bounded* if it takes the form $M_t = \mathbb{T}_{f,t}$ for $C$-bounded $f$. In particular, such $M$ have symmetric increments (even conditionally on the past) by definition of $\mathbb{T}_f$. We assume throughout that $C \geq 2$. The $\lesssim$ direction of equation 16 comes from the following immediate consequence of Freedman's inequality ((Freedman, 1975)).

**Lemma A.2.** *For any $C$-bounded martingale $M$ and $L, \gamma \geq 0$, we have*
$$\mathbb{P}\left[\exists t : \sup_{s \leq t} |M_t| \geq L\gamma, \ \langle M \rangle_t^{1/2} \leq L\right] \leq 2 \exp\left(-\frac{L\gamma^2}{2L + C\gamma}\right).$$

**Corollary A.3.** *For any set $\mathcal{M}$ of $C$-bounded martingales $M$, there is $\gamma = O(\log(Cn|\mathcal{M}|))$ so that*
$$\mathbb{P}\left[\exists (t, L) \in \mathbb{Z}_+ \times [C, \infty) : \sup_{M \in \mathcal{M}} \sup_{s \leq t} |M_t| \geq L\gamma, \ \sup_{M \in \mathcal{M}} \langle M \rangle_t^{1/2} \leq L\right] \leq (Cn)^{-2}.$$

*If $\mathcal{M}$ consists of $C$-integral martingales, then there is $\gamma = O(C \log(Cn|\mathcal{M}|))$ such that*
$$\mathbb{P}\left[\exists (t, L) \in \mathbb{Z}_+ \times (0, \infty) : \sup_{M \in \mathcal{M}} \sup_{s \leq t} |M_t| \geq L\gamma, \ \sup_{M \in \mathcal{M}} \langle M \rangle_t^{1/2} \leq L\right] \leq (Cn)^{-2}.$$

*Proof.* We union-bound over $M \in \mathcal{M}$ and $L = 2^k C$ for $0 \le k \le O(\log Cn)$, and apply Lemma A.2 with $\gamma$ replaced by $\gamma/2$ in each case. Note that by definition $\langle M \rangle_t^{1/2} \le C^2 n$ for all $t \le n$. In each case, since $L \ge C$ and $\gamma \ge 1$ we have

$$\frac{L\gamma^2}{2L + C\gamma} \gtrsim \min(\gamma^2, L\gamma/C) \ge \gamma.$$

Then we end up with the bound $O(\log Cn)|\mathcal{M}|e^{-\Omega(\gamma)}$ which gives the first desired estimate.

In the $C$-integral case, by definition $\langle M \rangle_t$ is a non-decreasing integer-valued process. Then it suffices to consider $L \ge 1$, so one may apply the first estimate with $\gamma' = C\gamma$. $\qquad\square$

The $\gtrsim$ direction of equation 16 follows from sub-exponential tails for the amount of quadratic variation a martingale can expend while staying in a strip. We emphasize that it is this step that requires a symmetric class $\mathcal{F}$, which ensures that taking an absolute value inside the supremum over the class never affects the value of the supremum.

**Lemma A.4.** *There exists an absolute constant $c > 0$ such that the following holds. For any discrete-time $C$-bounded martingale $M$ and any $L \ge C$, we have*

$$\mathbb{P}\left[ \exists t : \sup_{s \le t} |M_t| \le L, \ \langle M \rangle_t^{1/2} \ge L\gamma \right] \le 2\exp(-c\gamma^2).$$

*In the $C$-integral case, we also have for any $L > 0$:*

$$\mathbb{P}\left[ \exists t : \sup_{s \le t} |M_t| \le L, \ \langle M \rangle_t^{1/2} \ge CL\gamma \right] \le 2\exp(-c\gamma^2).$$

*Proof.* By monotone convergence, we can implicitly assume $M_t$ is defined only up to a finite time horizon $T$. Define $\tau_k$ to be the first time when $\langle M \rangle_{\tau_k} \ge 100L^2 k$ holds, or $\tau_k = T$ if this never occurs. Note that $M_t^2 - \langle M \rangle_t$ is a martingale with respect to a filtration we denote here by $(\mathcal{G}_t)$ (i.e. the filtration generated by the $\epsilon_t$ variables). Let $\gamma_k$ be the first time after $\tau_k$ that $|M_t| > L$, so that by $C$-boundedness $|M_{\gamma_k}| \le L + C \le 2L$, or $\gamma_k = \tau_{k+1}$ if this has not occured by time $\tau_{k+1}$. The optional stopping theorem gives

$$\mathbb{E}[M_{\gamma_k}^2 - \langle M \rangle_{\gamma_k}|\mathcal{G}_{\tau_k}] = M_{\tau_k}^2 - \langle M \rangle_{\tau_k}.$$

Assuming that $|M_{\tau_k}| \le L$, on the event that $\gamma_k = \tau_{k+1} < T$ the left-hand side is at most

$$4L^2 - \langle M \rangle_{\tau_k} - 90L^2 \le (M_{\tau_k}^2 - \langle M \rangle_{\tau_k}) - 80L^2.$$

On the other hand, if $|M_{\tau_k}| \le L$, then the left-hand side can never exceed $(M_{\tau_k}^2 - \langle M \rangle_{\tau_k}) + 10L^2$. We conclude that there is at most a $1/2$ probability that $\gamma_k = \tau_{k+1} < T$, conditionally on $\mathcal{G}_{\tau_k}$. By iteration, we conclude that

$$\mathbb{P}[\gamma_k = \tau_{k+1} < T \quad \forall k \le j] \le 2^{-j}.$$

This yields the first claimed estimate.

For $C$-integral martingales, as in Corollary A.3 it suffices to consider $L \ge 1$, so we again apply the first estimate with $\gamma' = C\gamma$. $\qquad\square$

**Corollary A.5.** *There exists an absolute constant $c > 0$ such that the following holds. For any set $\mathcal{M}$ of discrete-time $C$-bounded martingales $M$, we have*

$$\mathbb{P}\left[ \exists (t, L) \in \mathbb{Z}_+ \times [C, \infty) : \sup_{M \in \mathcal{M}} \sup_{s \le t} |M_t| \le L, \ \sup_{M \in \mathcal{M}} \langle M \rangle_t^{1/2} \ge L\gamma \right] \le 10\log(n)|\mathcal{M}|\exp(-c\gamma^2).$$

*In the $C$-integral case,*

$$\mathbb{P}\left[ \exists (t, L) \in \mathbb{Z}_+ \times (0, \infty) : \sup_{M \in \mathcal{M}} \sup_{s \le t} |M_t| \le L, \ \sup_{M \in \mathcal{M}} \langle M \rangle_t^{1/2} \ge CL\gamma \right] \le 10\log(n)|\mathcal{M}|\exp(-c\gamma^2).$$

*Proof.* For the first, we union-bound over $M \in \mathcal{M}$ and scales $L = C \cdot 2^k$, applying Lemma A.4 to each. The second is proved by again taking $\gamma' = C\gamma$. $\qquad\square$

Finally we give a maximal inequality which allows us to extract the endpoint value $\sup_{M \in \mathcal{M}} |M_n|$ which dictates the value of $\mathcal{R}_n^{\text{seq}}$.

**Lemma A.6.** *There is an absolute constant $c$ such that any finite set $\mathcal{M}$ of $C$-bounded martingales,*

$$\frac{c \, \mathbb{E} \sup_{M \in \mathcal{M}} \sup_{t \leq n} |M_t|}{\log(Cn)} - 1 \leq \mathbb{E} \sup_{M \in \mathcal{M}} |M_n| \leq \mathbb{E} \sup_{M \in \mathcal{M}} \sup_{t \leq n} |M_t|$$

*Furthermore, if all $M \in \mathcal{M}$ are $C$-integral, then the $-1$ term can be removed.*

*Proof.* The latter bound is trivial and holds deterministically without the expectation. To show the former we consider each dyadic scale $L = 2^k$ for $0 \leq k \leq O(\log(Cn))$. Letting

$$p_k = \mathbb{P}[\sup_{M \in \mathcal{M}} \sup_{t \leq n} |M_t| \geq 2^k]$$

we clearly have

$$c \cdot \mathbb{E} \sup_{M \in \mathcal{M}} \sup_{t \leq n} |M_t| \leq \sum_{0 \leq k \leq \log_2(Cn)} 2^k p_k.$$

Moreover we claim that

$$\mathbb{E} \sup_{M \in \mathcal{M}} |M_n| \geq \max_{k \in \mathbb{Z}} 2^k p_k \geq \left( \max_{0 \leq k \leq O(\log(Cn))} 2^k p_k \right) - 1.$$

Indeed the first estimate follows for each $k$ by letting $\tau_k$ be the first time that $|M_{\tau_k}| \geq 2^k$ holds for any $M \in \mathcal{M}$ and using the convexity of absolute value. The second estimate is clear. Comparing the two displays implies the result. Finally the extension to $C$-integral martingales is identical except that the term $-1$ in the previous display can be removed. $\qquad\square$

We now obtain a version of the contraction lemma with an additional $\text{poly}(\log(n|\mathcal{F}|))$ factor. This is the main step in the proof of Theorem 2.1; the remaining work is to discretize $\mathcal{F}$.

**Theorem A.7.** *For any $C$-bounded $\mathcal{F}$, any $\ell : \mathbb{R} \to \mathbb{R}$ satisfying $|\ell(x)| \leq |x|$ for all $x$, and any $\mathbf{z}$,*

$$n\mathcal{R}_n^{\text{seq}}(\ell \circ \mathcal{F}, \mathbf{z}) \leq O\big( \log^{3/2}(Cn|\mathcal{F}|) \log(Cn) \big) \cdot n\mathcal{R}_n^{\text{seq}}(\mathcal{F}, \mathbf{z}) + O(C \log(Cn|\mathcal{F}|)).$$

*Proof.* Let $\mathcal{M}$ consist of all $\mathbb{T}_f$ and $\widetilde{\mathcal{M}}$ consist of $\mathbb{T}_{\ell \circ f}$, for $f \in \mathcal{F}$. Symmetry of $\mathcal{F}$ means

$$\mathcal{R}_n^{\text{seq}}(\mathcal{F}, \mathbf{z}) = \mathbb{E} \sup_{M \in \mathcal{M}} |M_n|$$

while

$$\mathcal{R}_n^{\text{seq}}(\ell \circ \mathcal{F}, \mathbf{z}) = \mathbb{E} \sup_{\widetilde{M} \in \widetilde{\mathcal{M}}} \widetilde{M}_n \leq \mathbb{E} \sup_{\widetilde{M} \in \widetilde{\mathcal{M}}} |\widetilde{M}_n|.$$

We will relate these quantities using the comparison estimates above. By equation 17, almost surely:

$$\sup_{\widetilde{M} \in \widetilde{\mathcal{M}}} \langle \widetilde{M} \rangle_n \leq \sup_{M \in \mathcal{M}} \langle M \rangle_n.$$

Setting $\gamma_1 = O\big( \log(Cn|\mathcal{F}|) \big)$, Corollary A.3 holds with probability $(1 - (Cn)^{-2})$. Similarly taking $\gamma_2 = O\big( \sqrt{\log(Cn|\mathcal{F}|)} \big)$, we see that Corollary A.5 also holds with probability $(1 - (Cn)^{-2})$. Combining show that with probability $1 - 2(Cn)^{-2}$, **if**

$$\sup_{\widetilde{M} \in \widetilde{\mathcal{M}}} \sup_{t \leq n} |\widetilde{M}_t| \geq C\gamma_1, \tag{18}$$

**then** one may conclude

$$\begin{aligned}
\sup_{\widetilde{M} \in \widetilde{\mathcal{M}}} \sup_{t \leq n} |\widetilde{M}_t| &\leq \gamma_1 \sup_{\widetilde{M} \in \widetilde{\mathcal{M}}} \langle \widetilde{M} \rangle^{1/2} \\
&\leq \gamma_1 \sup_{M \in \mathcal{M}} \langle M \rangle^{1/2} \\
&\leq \gamma_1 \gamma_2 \sup_{M \in \mathcal{M}} \sup_{t \leq n} |M_t| \\
&= O\big( \log^{3/2}(Cn|\mathcal{F}|) \big) \sup_{M \in \mathcal{M}} \sup_{t \leq n} |M_t|.
\end{aligned}$$

Taking expectations, and accounting for the possibility that equation 18 does not hold, we conclude:

$$\mathbb{E} \sup_{M \in \mathcal{M}} \sup_{t \leq n} |\widetilde{M}_t| \leq O\big(\log^{3/2}(Cn|\mathcal{F}|)\big) \mathbb{E} \sup_{M \in \mathcal{M}} \sup_{t \leq n} |M_t| + O(C \log(Cn|\mathcal{F}|)).$$

By Lemma A.6, we pay another $\log(Cn)$ factor to remove the uniformity in times $t \leq n$. This implies the desired result. $\square$

**Theorem A.8.** *For any $C$-integral $\mathcal{F}$, any $\ell : \mathbb{R} \to \mathbb{R}$ satisfying $|\ell(x)| \leq |x|$ for all $x$, and any $\mathbf{z}$,*

$$n\mathcal{R}_n^{\mathrm{seq}}(\ell \circ \mathcal{F}, \mathbf{z}) \leq O\big(C^2 \log^{3/2}(Cn|\mathcal{F}|) \log(Cn)\big) \cdot n\mathcal{R}_n^{\mathrm{seq}}(\mathcal{F}, \mathbf{z}).$$

*Proof.* Let $\tau$ be the first time that some $f_* \in \mathcal{F}$ takes a non-zero value. By conditioning on $\epsilon_1, \ldots, \epsilon_\tau$, we may assume without loss of generality that $\tau = 0$. Then

$$\mathbb{E}[|\mathbb{T}_{f_*,n}(\mathbf{z})|] \geq \mathbb{E}[|\mathbb{T}_{f_*,1}(\mathbf{z})|] \geq 1 \tag{19}$$

by convexity of absolute value, and so $\mathcal{R}_n^{\mathrm{seq}}(\mathcal{F}, \mathbf{z}) \geq 1$.

The proof is now identical to Theorem A.7, but with $\gamma_1, \gamma_2$ each multiplied by $C$ and no additive terms, and no requirement of equation 18. Namely on the probability $1 - 2(Cn)^{-2}$ event that both Corollary A.3 and Corollary A.5 hold, we obtain

$$\sup_{\widetilde{M} \in \widetilde{\mathcal{M}}} \sup_{t \leq n} |\widetilde{M}_t| \leq O\big(C^2 \log^{3/2}(Cn|\mathcal{F}|)\big) \sup_{M \in \mathcal{M}} \sup_{t \leq n} |M_t|.$$

Then Lemma A.6 can be applied as before. The contribution from the remaining probability $\frac{2}{C^2 n^2}$ event is easily absorbed using equation 19, since $\max(|M_t|, |\widetilde{M}_t|) \leq Cn$ almost surely. $\square$

Finally we reduce to the finite $\mathcal{M}$ case. We recall (**?**)Definition 4]rakhlin2015sequential, specialized to the $\ell^\infty$ norm. It states that a set $V$ of $\mathbb{R}$-valued binary trees of depth $n$ is a sequential $\alpha$-cover for $\mathcal{F}$ if for any $f \in \mathcal{F}$ and any $\vec{\epsilon} \in \{\pm 1\}^n$, there exists $\mathbf{v} \in V$ with

$$|\mathbf{v}_t(\vec{\epsilon}) - f(\mathbf{z}_t(\vec{\epsilon}))| \leq \alpha, \quad \forall 1 \leq t \leq n.$$

We let the sequential covering number $\mathcal{N}_\infty(\alpha, \mathcal{F}, n)$ denote the minimal size of such a sequential cover. Then (**?**)Corollary 6]rakhlin2015sequential implies

$$\mathcal{N}_\infty(\alpha, \mathcal{F}, n) \leq (30Cn/\alpha)^d \tag{20}$$

for $d = \mathrm{sdim}_\alpha(\mathcal{F})$. In the integer-valued case, (**?**)Theorem 5]rakhlin2015sequential shows that $\mathcal{N}_\infty(1/2, \mathcal{F}, n) \leq (30Cn)^d$. Further, in this case the sequential cover can be taken to have equality $\mathbf{v}_t(\vec{\epsilon}) = f(\mathbf{z}_t(\vec{\epsilon}))$, by rounding $\mathbf{v}_t$ to the nearest integer. Using these tools, we now conclude the proof of Theorem 2.1.

*Proof of Theorem 2.1.* The $C$-integral case is immediate from Theorem A.7 with $|\mathcal{F}|$ replaced by $(30Cn)^d$, since then $\log |\mathcal{F}| \leq O(d \log(Cn))$.

In the $C$-regular case, we discretize $[-C, C]$ into an $\alpha$-net via intersection with $\alpha\mathbb{Z}$. We then round each $f \in \mathcal{F}$ or $\ell \circ \mathcal{F}$ to the nearest element of this set, obtaining $\hat{\mathcal{F}}$ and $\widehat{\ell \circ \mathcal{F}}$. Then the Hölder continuity assumption implies that $\mathcal{R}_n^{\mathrm{seq}}(\mathcal{F})$ and $\mathcal{R}_n^{\mathrm{seq}}(\ell \circ \mathcal{F})$ are affected by $\alpha$ and $\delta$ respectively. The first estimate in equation 15 is obtained by noting that up to scaling by $\alpha$, the rounded function class is simply a $C'$-integral class for $C' \leq O(C/\alpha)$. Thus

$$n\mathcal{R}_n^{\mathrm{seq}}(\widehat{\ell \circ \mathcal{F}}, \mathbf{z}) \leq O\big((C/\alpha)^2 \log^{3/2}(Cn|\hat{\mathcal{F}}|/\alpha) \log(Cn/\alpha)\big) \cdot n\mathcal{R}_n^{\mathrm{seq}}(\hat{\mathcal{F}}, \mathbf{z}).$$

Combining with equation 20 gives the first estimate in equation 15.

The second estimate in equation 15 is similarly obtained using equation 20, but now by applying Theorem A.7 (and thus treating $\hat{\mathcal{F}}$ and $\widehat{\ell \circ \mathcal{F}}$ as general $C$-bounded classes). $\square$

# B. Proof of Lemma 3.3

Lemma 3.3 follows from known results in stochastic processes. Indeed a variant of Donsker's celebrated functional central limit theorem for this process is shown in (Amir, 1991). Namely, consider the random function $f_n : [0, 1] \to \mathbb{R}$ defined by

$$f_n(m/n) = n^{-1/2} a_m, \quad \forall m \in \{0, 1, \ldots, n\}$$

and by piece-wise linear interpolation for other inputs. Then $f_n$ converges in distribution to a non-trivial Markov process on $\mathbb{R}_+$ known as sticky (reflected) Brownian motion. In particular, one immediately finds that the limiting law for $a_n/\sqrt{n}$ is nontrivial and supported on $[0, \infty)$, hence $\mathbb{E}[a_n] \geq \Omega(\sqrt{n})$.

For the upper bound, notice that there exists a coupling to an ordinary simple random walk $x_0, x_1, \ldots, x_n$ under which $a_n = |x_m|$ for some random $0 \leq m \leq n$ (namely the number of times $t$ for which $a_t \neq a_{t+1}$). It is a well-known consequence of the reflection principle that $\mathbb{E}[\max_{0 \leq m \leq n} |x_m|] \leq O(\sqrt{n})$. This proves the matching upper bound and completes the proof.

# C. Proof of Theorem 2.3

In order to prove the first statement in Theorem 2.3, we must proceed differently from the proof of Theorem 2.2 in order to enforce symmetry of $\mathcal{F}$. Indeed, we modify the domain $\mathcal{Z}$ from that proof such that an element $\mathbf{z} \in Z$ takes the form $\mathbf{z} = (y, \sigma) \in \mathbb{Z}_+ \times \{1, -2\} = \mathcal{Z}$. We now consider functions of the form

$$f_i((y, \sigma)) = \sigma \cdot 1_{y=i}, \quad i \in \mathbb{Z}_{\geq 0}. \tag{21}$$

In other words for each $i \geq 0$, the function $f_i$ takes value 1 on $\mathbf{z} = (i, 1)$, value $-2$ on $\mathbf{z} = (i, -2)$, and is otherwise zero. (And $f_0 \equiv 0$.) This choice of $\mathcal{F}$ still has $d = 1$ for the same reason as before. We take $\ell$ to be the clipping function

$$\ell(x) = \text{sign}(x) \cdot \min(1, |x|).$$

The intuition now is that if the partial sums for $\mathbb{T}_f$ evolve in a "cycle" $0 \to 1 \to 2 \to 0$, then the corresponding partial sums for $\mathbb{T}_{\ell \circ f}$ evolve as $0 \to 1 \to 2 \to 1$. If this cycle repeats many times, then we will have $|\mathbb{T}_{\ell \circ f}| \gg |\mathbb{T}_f|$, as the former diverges linearly while the latter remains bounded. As in the proof of Theorem 2.2, we can ensure that $\mathcal{R}_n^{\text{seq}}(\mathcal{F}) \leq O(1)$ by moving to the next $f \in \mathcal{F}$ whenever this cycle breaks, thus getting many attempts at a long cycle. The resulting lower bound of $\log n$ comes from the typical length of the longest constant block in a length $n$ binary string, which is the content of the following lemma.

**Lemma C.1.** *Let $\epsilon \in \{\pm 1\}^n$ be uniformly random. Then for $L = 0.1 \cdot \log n$, with probability at least $1/2$ there exists a substring $(\epsilon_t, \epsilon_{t+1}, \ldots, \epsilon_{t+L-1}) = (1, 1, \ldots, 1)$ of $L$ consecutive 1's in $\epsilon$.*

*Proof.* Simply consider all $t$ that are multiples of $\lfloor \sqrt{n} \rfloor$, making each of the $\sqrt{n}$ events $\mathcal{E}_t = \{\epsilon_t = \epsilon_{t+1} = \cdots \epsilon_{t+L-1} = 1\}$ independent. For each $t$, $\mathbb{P}(\mathcal{E}_t) = 2^{-L} \geq 1/\sqrt{n}$, so the claim follows from the fact that $1 - \left(1 - \frac{1}{10\sqrt{n}}\right)^{\sqrt{n}} \geq 1/2$. $\square$

We are now ready to prove the first statement of Theorem 2.3.

*Proof of Theorem 2.3, first statement:* We take $\mathcal{F}$ to consist of the functions in equation 21 and their negations. Define the periodic sequence

$$S = (s_0, s_1, s_2, \ldots) = (0, 1, 2, 0, 1, 2, \ldots),$$

i.e. $s_i = 3\{i/3\}$ where $\{\cdot\}$ denotes fractional part. Define also

$$\widetilde{S} = (\widetilde{s}_0, \widetilde{s}_1, \widetilde{s}_2, \ldots) = (0, 1, 2, 1, 2, 3, 2, 3, 4, \ldots),$$

i.e. $\widetilde{s}_i = 3\{i/3\} + \lfloor \frac{i}{3} \rfloor$.

At depth $t$, we define $\sigma_t = -2$ if $t$ is a multiple of 3 and otherwise $\sigma_t = 1$. We always have $y_{3s} = y_{3s+1} = y_{3s+2}$ and set $y_{3s+3} = y_{3s}$ iff $\epsilon_{3s} = \epsilon_{3s+1} = \epsilon_{3s+2} = 1$. Otherwise we set $y_{3s+3} = y_{3s} + 1$. It is easy to see that $\mathbb{T}_{f_i}$ follows $S$ during the time interval where $y_t = i$, except for $O(1)$ deviations at the end. This implies the first part of equation 8 since $S$ is uniformly bounded.

Moreover, if $\epsilon_t = 1$ for all $t \in \left[3s, 3(s+L)\right) \cap \mathbb{Z}$ then

$$\mathbb{T}_{\ell \circ f_{y_{3s}}, 3(s+L)} = L.$$

Indeed, the sequence $t \mapsto \mathbb{T}_{\ell \circ f_{y_{3s}}, 3s+t}$ exactly follows $\widetilde{S}$ as defined above in this case. In particular we find that

$$\mathbb{T}_{\ell \circ f_{y_{3s}}, n} \geq L - O(1).$$

Then the second part of equation 8 follows from Lemma C.1 as desired. $\qquad\square$

For larger $d$, one obtains an $\Omega(d)$ lower bound by implementing the above construction "in parallel" rather than "in series". Our construction works for any $d \leq n$ and satisfies $|\mathcal{F}| = 2^d$; this always implies $\mathrm{Ldim}\,\mathcal{F} \leq d$, with equality holding in this case. In fact one can easily reduce to the case $n = d$ as explained at the start of the proof below. For this case, one takes

$$f_\alpha(\epsilon_1, \ldots, \epsilon_{t-1}) = \alpha_i \sigma_t \cdot 1_{(\epsilon_1, \ldots, \epsilon_{t-1}) = (\alpha_1, \ldots, \alpha_{t-1})}.$$

The point is to extend the previous construction to happen in parallel such that the $2^d$ functions in $\mathcal{F}$ correspond bijectively with $\epsilon \in \{\pm 1\}^d$ (we note that $(\epsilon_{d+1}, \ldots, \epsilon_n)$ are irrelevant since $\mathbf{z}_t = 0$ for $t > d$).

*Proof of Theorem 2.3, second statement:* First we reduce to the case $n = d$. If $n$ is strictly larger, we simply make nothing happen on times $t > d$. This is done by augmenting $Z$ with a new element $\mathbf{z}_* \in Z$ such that $f(\mathbf{z}_*) = 0$, which changes $\mathrm{Ldim}(\mathcal{F})$ by at most 1. Thus we assume $n = d$ below.

We will index $\mathcal{F}$ by $\alpha \in \{\pm 1\}^d$; hence for each $\alpha \in \{\pm 1\}^d$ there will be $f_\alpha \in \mathcal{F}$ such that

$$\mathbb{T}_{f_\epsilon, t}(\epsilon) = S_t, \quad t \in [d]; \qquad \mathbb{T}_{\ell \circ f_\epsilon, t}(\epsilon) = \widetilde{S}_t, \quad t \in [d]. \tag{22}$$

Since the bound $\mathrm{Ldim}(\mathcal{F}) \leq \log_2(|\mathcal{F}|) = d$ holds in general, we do not need to worry over the structure of the set $Z$. Thus we let each $\mathbf{z}_t(\epsilon_1, \ldots, \epsilon_{t-1}) = (\epsilon_1, \ldots, \epsilon_{t-1})$ simply denote its entire history. With $\mathbf{z}$ fixed in this way, we can just ignore $\mathbf{z}$ and define each $f_\alpha(\epsilon_1, \ldots, \epsilon_{t-1}) = f_\alpha(\mathbf{z}_t)$ directly. Set

$$f_\alpha(\epsilon_1, \ldots, \epsilon_{t-1}) = \alpha_i \sigma_t \cdot 1_{(\epsilon_1, \ldots, \epsilon_{t-1}) = (\alpha_1, \ldots, \alpha_{t-1})}.$$

Here as in the above proof of the first statement of Theorem 2.3, we let $\sigma_t = -2$ if $t$ is a multiple of 3 and otherwise $\sigma_t = 1$. It is easy to see that equation 22 holds. Also, once $\alpha$ and $\epsilon$ differ the partial sums do not change. This finishes the proof since $\mathbb{T}_{\ell \circ f_\epsilon, d} = \widetilde{S}_d = d/3 \pm O(1)$ while $\mathbb{T}_{f_\alpha, d}$ is uniformly bounded. $\qquad\square$

# D. Offset Rademacher Upper bound

In this section, we provide the complete proof of Theorem 2.4. As stated in the main body, the structure of the proof is to relate the offset Rademacher complexity to the critical radius $\tilde{r}_n(\mathcal{F})$ defined in Definition 4.1 as in equation 14, i.e., show that

$$\frac{n\tilde{r}_n(\mathcal{F})^2}{\mathrm{polylog}(n)} \lesssim \mathcal{R}_n^\eta(\mathcal{F}) \lesssim n\tilde{r}_n(\mathcal{F})^2$$

whenever $\eta$ and $\mathcal{F}$ satisfy the conditions of the theorem. The result will then follow from the fact that $\tilde{r}_n(\mathcal{F})$ satisfies contraction, phrased as Lemma 4.2 and proved now.

*Proof of Lemma 4.2.* Recall that by definition,

$$\tilde{r}_n(\mathcal{F}) = \sup \left\{ r \mid nr^2 + 2\log(2) \leq \log \mathcal{N}\left(\mathcal{F}, \|\cdot\|_n, r\right) \right\}.$$

We observe that by the definition of a Lipschitz function, an $\epsilon$-net on $\mathcal{F}$ induces an $\epsilon$-net on $\phi \circ \mathcal{F}$ and thus $\mathcal{N}\left(\phi \circ \mathcal{F}, \|\cdot\|_n, r\right) \leq \mathcal{N}\left(\mathcal{F}, \|\cdot\|_n, r\right)$. Thus it holds that for any $r \leq \tilde{r}_n(\mathcal{F})$,

$$nr^2 + 2\log(2) \leq \log \mathcal{N}\left(\phi \circ \mathcal{F}, \|\cdot\|_n, r\right) \leq \log \mathcal{N}\left(\mathcal{F}, \|\cdot\|_n, r\right),$$

which concludes the proof. $\qquad\square$

We divide the proof into the lower and upper bounds in equation 14 with the upper bound proof appearing in Appendix D.1 and the lower bound proof appearing in Appendix D.2. We then conclude the proof by showing that the critical radius satisfies contraction in Appendix D.3

### D.1. Upper Bound on Offset Rademacher Complexity

We begin by showing the upper bound, which holds for any Donsker function class $\mathcal{F}$. To do this, we denote the localized Dudley integral by

$$\mathcal{J}\left(\mathcal{F}, \|\cdot\|_n, \gamma\right) = \int_0^\gamma \sqrt{\log \mathcal{N}\left(\mathcal{F}, \|\cdot\|_n, \delta\right)} d\delta$$

and we recall the following result controlling the offset Rademacher complexity in terms of the Dudley integral:

**Lemma D.1** (Lemma 6 from Liang et al. (2015)). *For any class of functions $\mathcal{F} : \mathcal{X} \to \mathbb{R}$, and any data set $x_1, \ldots, x_n$, it holds that*

$$n \cdot \mathcal{R}_n^\eta(\mathcal{F}) \leq \inf_{\substack{\gamma > 0 \\ \alpha \leq \gamma}} \left\{ \frac{2 \log \left(\mathcal{N}(\mathcal{F}, \|\cdot\|_n, \gamma)\right)}{\eta} + 4\alpha n + 12\sqrt{n} \cdot \int_\alpha^\gamma \sqrt{\log \mathcal{N}\left(\mathcal{F}, \|\cdot\|_n, \delta\right)} d\delta \right\}.$$

We can now prove the following upper bound on the offset Rademacher complexity.

**Lemma D.2.** *Suppose that $\mathcal{F} : \mathcal{X} \to \mathbb{R}$ is a function class such and $\mathcal{N}$ is an upper bound on the log-covering number of $\mathcal{F}$ as in Definition 4.1 with associated critical radius $\tilde{r}_n(\mathcal{F})$. Suppose that $\mathcal{N}$ satisfies the following two properties:*

1. *There exists some $C > 0$ such that for $0 < \gamma \leq \tilde{r}_n(\mathcal{F})$, it holds that the Dudley integral is upper bounded as $\mathcal{J}\left(\mathcal{F}, \|\cdot\|_n, \gamma\right) \leq C\gamma\sqrt{\mathcal{N}(\gamma)}$.*

2. *For all $\gamma \leq \tilde{r}_n(\mathcal{F})$, it holds that $\mathcal{N}(\gamma) \geq 5$.*

*Then it holds that*

$$\mathcal{R}_n^\eta(\mathcal{F}) \leq \left(\frac{2}{\eta} + 24\right) \tilde{r}_n(\mathcal{F})^2.$$

*Proof.* We begin by noting that by Lemma D.1, setting $\alpha = 0$, it holds that for any $\gamma \geq 0$

$$n \cdot \mathcal{R}_n^\eta(\mathcal{F}) \leq \frac{2 \log \mathcal{N}\left(\mathcal{F}, \|\cdot\|_n, \gamma\right)}{\eta} + 12\sqrt{n} \cdot \mathcal{J}\left(\mathcal{F}, \|\cdot\|_n, \gamma\right)$$

$$\leq \frac{2 \log \mathcal{N}(\gamma)}{\eta} + 12\gamma\sqrt{n\mathcal{N}(\gamma)},$$

where we used the assumption on the Dudley integral to get the second inequality. Suppose that we choose $\gamma = \tilde{r}_n(\mathcal{F})$. We now observe that right continuity of $\gamma \mapsto \mathcal{N}(\gamma)$ ensures that

$$\mathcal{N}(\tilde{r}_n(\mathcal{F})) \leq n\tilde{r}_n(\mathcal{F})^2 + 2 \log 2.$$

Indeed, for any $\epsilon > 0$, it holds that

$$n\left(\tilde{r}_n(\mathcal{F}) + \epsilon\right)^2 + 2\log(2) > \mathcal{N}(\tilde{r}_n(\mathcal{F}) + \epsilon).$$

Applying right-continuity of both sides of the inequality proves the claim. Thus it holds that

$$n \cdot \mathcal{R}_n^\eta(\mathcal{F}) \leq \frac{2}{\eta}\left(n\tilde{r}_n(\mathcal{F})^2 + 2\log(2)\right) + 12\tilde{r}_n(\mathcal{F})\sqrt{n\left(n\tilde{r}_n(\mathcal{F})^2 + 2\log(2)\right)}$$

$$= \left(\frac{2}{\eta} + 12\right) n\tilde{r}_n(\mathcal{F})^2 + 12\sqrt{2\log(2)n\tilde{r}_n(\mathcal{F})^2}.$$

Observing that by the second condition on $\mathcal{F}$, we have $\mathcal{N}(\tilde{r}_n(\mathcal{F})) \geq 5 \geq 4\log(2)$ and so we see that

$$\tilde{r}_n(\mathcal{F})^2 \geq \frac{2\log(2)}{n} \tag{23}$$

and thus

$$\sqrt{n\tilde{r}_n(\mathcal{F})^2} \leq n\tilde{r}_n(\mathcal{F})^2.$$

Putting everything together concludes the proof. □

## D.2. Lower Bound on Offset Rademacher Complexity

We are now ready to prove the lower bound. It is here where convexity of $\mathcal{F}$ is necessary; the complexity assumption on $\mathcal{F}$ is superfluous however.

**Lemma D.3.** *Suppose that $\mathcal{F} : \mathcal{X} \to [-1, 1]$ is a convex function class. Suppose that $\gamma \mapsto \mathcal{N}(\gamma)$ is as in Definition 4.1 and suppose that $\mathcal{M}(\epsilon)$ is a right-continuous, non-decreasing lower bound on the packing number of $\mathcal{F}$. Further suppose that there is a constant $C \geq 1$ that for all $\tilde{r}_n(\mathcal{F}) \geq \gamma > 0$,*

$$\mathcal{N}(\gamma) \leq C\mathcal{M}\left(\frac{\gamma}{C}\right) \qquad\qquad \mathcal{M}(\gamma) \leq C^2\mathcal{M}\left(C\gamma\right). \qquad (24)$$

*Then it holds for any $\eta \leq (4\sqrt{\log(n)})^{-1}$ that*

$$\mathcal{R}_n^{\eta}(\mathcal{F}) \geq \frac{\tilde{r}_n(\mathcal{F})^2}{2^{18} \cdot C^4 \sqrt{\log(n)}}.$$

As we will see in Lemma D.7, the assumptions in equation 24 are satisfied for convex, Donsker $\mathcal{F}$. Our proof of Lemma D.3 proceeds by first recalling that the offset Rademacher complexity upper bounds the estimation error of Empirical Risk Minimization (ERM) for fixed design regression over convex $\mathcal{F}$ with bounded noise, then applying the standard Fano' lower bound to show that this problem with Gaussian noise cannot achieve prediction error better than $\tilde{r}_n(\mathcal{F})^2$, and concluding with a truncation argument. To begin, we recall:

**Theorem D.4** (Theorem 3 from (Liang et al., 2015)). *Suppose that $\mathcal{F} : \mathcal{X} \to [-1, 1]$ is convex and symmetric, $X_1, \ldots, X_n \in \mathcal{X}$, and $Y_i = f^{\star}(X_i) + \xi_i$ with $\xi_i$ independent and identically distributed mean zero random variables. Suppose that*

$$\hat{f} \in \arg\min_{f \in \mathcal{F}} \sum_{i=1}^{n} (f(X_i) - Y_i)^2$$

*is the ERM estimator. If $|Y - f(X)| \leq M$ almost surely with $M \geq 3 \vee \eta^{-1}$ and $\eta \leq 1$, then*

$$\mathbb{E}\left[(\hat{f}(X) - Y)^2\right] - \mathbb{E}\left[(f^{\star}(X) - Y)^2\right] \leq \left(4M + \frac{3}{2}\right)\mathcal{R}_n^{\eta}(\mathcal{F}),$$

*where $\eta = \frac{1}{4M}$.*

By the centred assumption of the $\xi_i$, we have

$$\mathbb{E}\left[\left(\hat{f}(X) - Y\right)^2\right] - \mathbb{E}\left[(f^{\star}(X) - Y)^2\right] = \mathbb{E}\left[(\hat{f}(X) - f^{\star}(X))^2\right] \qquad (25)$$

and thus the offset Rademacher complexity is bounded below by the performance of the ERM predictor, assuming that $\mathcal{F}$ is convex. Note that convexity of $\mathcal{F}$ is critical here, as (**?**)Theorem 3]liang2015learning is proved for the performance of the star estimator, which reduces to the ERM for a convex function class; for nonconvex classes, this bound *does not* hold .

We now prove the classical lower bound for Gaussian regression derived from Fano's inequality. We follow the approach of (Yang & Barron, 1999). We recall the following result:

**Theorem D.5** (Theorem 1 from (Yang & Barron, 1999)). *Supppose that $d$ is a distance satisfying the triangle inequality. Define the quantities*

$$\epsilon_n = \sup\left\{\epsilon > 0 | n\epsilon^2 \leq \mathcal{N}(\epsilon)\right\}$$
$$\underline{\epsilon}_n = \sup\left\{\epsilon > 0 | 4n\epsilon_n^2 + 2\log 2 \leq \mathcal{M}(\epsilon)\right\}.$$

*Suppose that $(X_i, Y_i) \sim P_f$ for some probability distribution parameterized by $f \in \mathcal{F}$. Then,*

$$\inf_{\hat{f}} \sup_{f^{\star} \in \mathcal{F}} \mathbb{E}_{f^{\star}}\left[d(\hat{f}, f^{\star})^2\right] \geq \frac{\epsilon_n^2}{8}$$

*where the infimum is over all maps from the data set to $\mathcal{F}$.*

We have the following lower bound:

**Lemma D.6.** *Suppose that $\mathcal{F} : \mathcal{X} \to \mathbb{R}$ is a convex, symmetric function class and $X_1, \ldots, X_n \in \mathcal{X}$ are given. Let $\gamma_1, \ldots, \gamma_n$ be independent standard Gaussians and suppose that $Y_i = f^\star(X_i) + \gamma_i$ for some $f^\star \in \mathcal{F}$. Let $\mathcal{N}, \mathcal{M}$ be as in Lemma D.3. Then the following inequality holds:*

$$\inf_{\hat{f}} \sup_{f^\star \in \mathcal{F}} \mathbb{E}\left[ \left\| \hat{f} - f^\star \right\|_n^2 \right] \geq \frac{\tilde{r}_n(\mathcal{F})^2}{8192 C^4},$$

*where the infimum is over all random $\hat{f} \in \mathcal{F}$ measurable with respect to $X_1, \ldots, X_n, Y_1, \ldots, Y_n$.*

*Proof.* We let $d^2(f, f') = \frac{1}{n} \cdot D_{\text{KL}}\left( P_{Y|X,f} \big\| P_{Y|X,f'} \right)$. By the assumption of Gaussianity, we see that

$$d^2(f, f') = \frac{1}{2} \| f - f' \|_n^2$$

and so $d$ satisfies the triangle inequality. Thus we may apply Theorem D.5 and observe that

$$\inf_{\hat{f}} \sup_{f^\star} \mathbb{E}\left[ \left\| \hat{f} - f^\star \right\|_n^2 \right] \geq \frac{\epsilon_n^2}{8}.$$

We now need to bound $\underline{\epsilon}_n$ by $\tilde{r}_n(\mathcal{F})$. To do this we begin by observing that by equation 23, we have

$$\tilde{r}_n(\mathcal{F})^2 \geq \frac{2 \log(2)}{n}$$

and so

$$4n\tilde{r}_n(\mathcal{F})^2 \geq n\tilde{r}_n(\mathcal{F})^2 + 2\log(2) \geq \mathcal{N}(\tilde{r}_n(\mathcal{F})) \geq \mathcal{N}(2\tilde{r}_n(\mathcal{F})).$$

This implies that $\epsilon_n \leq 2\tilde{r}_n(\mathcal{F})$. We now observe that

$$4n\epsilon_n^2 + 2\log(2) \leq 16n\tilde{r}_n(\mathcal{F})^2 + 2\log 2 \leq 32n\tilde{r}_n(\mathcal{F})^2,$$

where we again used equation 23. By construction of $\tilde{r}_n(\mathcal{F})$, we see that

$$32n\tilde{r}_n(\mathcal{F})^2 = 128n \left( \frac{\tilde{r}_n(\mathcal{F})}{2} \right)^2 \leq 128\mathcal{N}\left( \frac{\tilde{r}_n(\mathcal{F})}{2} \right).$$

Now, applying the inequalities of equation 24, we see that

$$128\mathcal{N}\left( \frac{\tilde{r}_n(\mathcal{F})}{2} \right) \leq 128C\mathcal{M}\left( \frac{\tilde{r}_n(\mathcal{F})}{2C} \right) \leq \mathcal{M}\left( \frac{\tilde{r}_n(\mathcal{F})}{32C^2} \right).$$

Combining the preceding inequalities, we see that

$$\mathcal{M}\left( \frac{\tilde{r}_n(\mathcal{F})}{32C^2} \right) \geq 4n\epsilon_n^2 + 2\log(2).$$

Thus, by construction, we obtain:

$$\underline{\epsilon}_n \geq \frac{\tilde{r}_n(\mathcal{F})}{32C^2}.$$

The result follows. $\qquad\square$

Finally, we apply a truncation argument to show that, with a correctly tuned $M$ in Theorem D.4, we can lower bound $\mathcal{R}_n^\eta(\mathcal{F})$ by $\tilde{r}_n(\mathcal{F})$.

*Proof of Lemma D.3.* Let $\xi_i \sim N(0,1)$ be standard Gaussians and for any $M > 0$, let $\xi_i^M = \xi_i \mathbb{I}[|\xi_i| \le M]$ be the same Gaussians truncated at level $M$. By the Gaussian maximal inequality, with probability at least $1 - \delta$ it holds that

$$\max_{1 \le i \le n} |\xi_i| \le \sqrt{2 \log\left(\frac{2n}{\delta}\right)}.$$

Let $\mathcal{A}_M$ denote the event that $\xi_i = \xi_i^M$ for all $1 \le i \le M$. By construction, then, if $M \ge \sqrt{2 \log\left(\frac{2n}{\delta}\right)}$, it holds that $\mathbb{P}(\mathcal{A}_M) \ge 1 - \delta$. Let $Y_i = f^\star(X_i) + \xi_i$ and let $Y_i^M = f^\star(X_i) + \xi_i^M$. Let $\hat{f}$ denote the ERM evaluated on the data $(X_i, Y_i)$ and let $\hat{f}^M$ denote the ERM evaluated on the data $(X_i, Y_i^M)$. By Theorem D.4 and Lemma D.6 and equation 25, for any $M \le 3 \vee \eta^{-1}$, we compute:

$$
\begin{aligned}
\mathcal{R}_n^\eta(\mathcal{F}) &\ge \frac{2}{4M+3} \cdot \mathbb{E}\left[\left\|\hat{f}^M - f\right\|_n^2\right] \\
&= \frac{2}{4M+3} \cdot \left(\mathbb{E}\left[\left\|\hat{f} - f\right\|_n^2\right] - \mathbb{E}\left[\left\|\hat{f} - f\right\|_n^2 \mathbb{I}[\mathcal{A}_M^c]\right]\right) \\
&\ge \frac{2}{4M+3} \left(\frac{\tilde{r}_n(\mathcal{F})^2}{8192 C^4} - 4 \cdot \mathbb{P}(\mathcal{A}_M^c)\right)
\end{aligned}
$$

Thus, if we choose $M = 16\sqrt{\log(n)}$ we see that

$$\mathcal{R}_n^\eta(\mathcal{F}) \ge \frac{1}{32\sqrt{\log(n)}} \cdot \frac{\tilde{r}_n(\mathcal{F})^2}{8192 C^4},$$

where we once again applied equation 23. The result follows. $\qquad \square$

### D.3. Conclusion of Proof

If $\mathcal{F}$ is a singleton then the result is trivial. Otherwise, note that log-covering numbers grow at least logarithmically for convex function classes with at least 2 elements by Lemma D.7. By the assumption that $\mathcal{F}$ is Donsker, we may choose $\mathcal{N}$ such that

$$\log\left(\frac{1}{\epsilon}\right) \le \log \mathcal{N}(\epsilon) \le \epsilon^{-2}.$$

Thus we may choose $\mathcal{N}$ and $\mathcal{M}$ such that the conditions of Lemma D.3 are satisfied. Indeed, we may choose $\tilde{r}_n(\mathcal{F})^2 \lesssim n^{-2/(2+p)}$ whenever $\log \mathcal{N}(\mathcal{F}, \|\cdot\|_n, \epsilon) \lesssim \epsilon^{-p}$ for some $p > 0$ and thus we may take $\mathcal{N}(\tilde{r}_n(\mathcal{F})) \ge 5$ for $n$ sufficiently large.

Combining Lemmas 4.2, D.2 and D.3, we compute:

$$
\begin{aligned}
\mathcal{R}_n^\eta(\phi \circ \mathcal{F}) &\le \left(\frac{2}{\eta} + 24\right) n \cdot \tilde{r}_n(\phi \circ \mathcal{F})^2 \\
&\le \left(\frac{2}{\eta} + 24\right) n \cdot \tilde{r}_n(\mathcal{F})^2 \\
&= \left(\frac{2}{\eta} + 24\right) \cdot 2^{18} \cdot C^4 \sqrt{\log(n)} \cdot \frac{n\tilde{r}_n(\mathcal{F})^2}{2^{18} \cdot C^4 \sqrt{\log(n)}} \\
&\le \left(\frac{2}{\eta} + 24\right) \cdot 2^{18} \cdot C^4 \sqrt{\log(n)} \cdot \mathcal{R}_n^\eta(\mathcal{F}).
\end{aligned}
$$

The result follows. All that remains is to prove the following lemma.

**Lemma D.7.** *Let $\mathcal{F} : \mathcal{X} \to \mathbb{R}$ be a convex function class with at least two elements. Then for sufficiently small $\epsilon$ it holds that $\log \mathcal{N}(\mathcal{F}, \|\cdot\|_n, \epsilon) \gtrsim \log(1/\epsilon)$.*

*Proof.* Let $f_0, f_1 \in \mathcal{F}$ such that $\|f_0 - f_1\|_n = \delta > 0$ for some $\delta$; such functions exist by the assumption of $\mathcal{F}$ having at least two elements. Let $f_\lambda = (1 - \lambda)f_0 + \lambda f_1$ and note that convexity requires $f_\lambda \in \mathcal{F}$. Simple computation ensures that for any $\lambda, \lambda'$,

$$\|f_\lambda - f_{\lambda'}\|_n^2 = (\lambda - \lambda')^2 \delta^2.$$

Thus $\{f_{i\epsilon} | i \in \mathbb{N} \text{ and } i \leq 1/\epsilon\}$ forms an $(\epsilon\delta)$-packing of $\mathcal{F}$. $\qquad \square$

# E. Proof of Theorem 2.6

In the offset case, the intuition for the construction is quite different. Observe that for $\eta > 0$ not too close to $0$, and $\mathcal{F}$ as in equation 11, one expects only the small values of $t$ to be relevant in maximizing equation 5. If most of $\mathcal{F}$ is too large to help maximize equation 5, then we could use a contraction $\ell$ to "fold" the large parts of $\mathcal{F}$ in towards the origin, creating a much richer function class at the relevant (small) scale for equation 5. Our analysis identifies a class of "almost piece-wise constant" functions, such that any $n^{\Omega(1)}$-sized collection of these functions can be simultaneously contained in $\ell \circ \mathcal{F}$ (by judicious choice of $\ell$). Then we show that a *random* subset of almost piece-wise constant functions suffices to increase the offset Rademacher complexity by a factor $\Omega(\log n)$. The idea is that for typical $\vec{\epsilon}$, the losses of each function are conditionally independent and can be individually understood via the (Berry-Esseen bound for the) central limit theorem.

## E.1. Proof Outline

The proof details for Theorem 2.6 are a bit technical so we begin with a brief outline.

First, we take $\eta = n^{-c}$ for a very small constant $c$. We take $\mathcal{F}$ as in equation 11 with

$$f_*(x_i) = 1 - in^{-1.01}. \tag{26}$$

Our first insight is that $f_*$ and $\vec{\epsilon}$ will typically have correlation $O(n^{-1/2}) \ll 1$. Hence the optimal $f \in \mathcal{F}$ will be $tf_*$ for $t \leq \tilde{O}(n^{-1/2})$ quite small. This means that the vast majority of the class $\mathcal{F}$ cannnot contribute to the supremum, in contrast with the $\eta = 0$ case where only extreme points of $\mathcal{F}$ matter.

Since most functions in $\mathcal{F}$ are too large to maximize equation 5, we use the Lipschitz $\ell$ to "fold" the large parts of $\mathcal{F}$ in towards the origin, creating a much richer function class at the relevant (small) scale for equation 5. Our analysis implements this idea using a class of "almost piece-wise constant" functions. In Proposition E.4 we show that it is possible to force any $\Omega(n^{1/1000})$ such functions to lie in $\ell \circ \mathcal{F}$ (by judicious choice of $\ell$). This leads to the $\Omega(\log n)$ bound from the tail decay of a Gaussian. (Indeed a generic function class of size $k$ will have offset Rademacher complexity proportional to $\frac{\log k}{n}$ rather than $\sqrt{\frac{\log k}{n}}$.) Of course, the correlations of $\vec{\epsilon}$ with the different functions in $\ell \circ \mathcal{F}$ are neither exactly independent nor Gaussian, so care is needed even in the later stages. We will in fact see that a *random* choice of almost piece-wise constant functions works well on average.

## E.2. Technical Details

We first verify the behavior of $\mathcal{R}_n^\eta$ is as above.

**Proposition E.1.** *Fix* $f_* : X \to [-1, 1]$ *(for instance as in equation 26) and let* $\mathcal{F}$ *be as in equation 11. Then* $\mathcal{R}_n^\eta(\mathcal{F}) \leq \frac{1}{2n\eta}$.

*Proof.* Let $S = \frac{1}{n} \sum_{i=1}^n \epsilon_i f_*(x_i)$. Then it is easy to see that

$$\mathcal{R}_n^\eta(\mathcal{F}; \mathbf{x}, \vec{\epsilon}) = \max_{t \in [-1,1]} \left(tS - \eta t^2\right) \leq \max_{t \in \mathbb{R}} \left(tS - \eta t^2\right) = \frac{S^2}{2\eta}.$$

Since $\mathbb{E}[S^2] \leq 1/n$, we find that $\mathcal{R}_n^\eta(\mathcal{F}) \leq \frac{1}{2n\eta}$. $\qquad \square$

We next turn to counterexamples. Fixing $f_* : X \to [-1, 1]$ as in equation 26, the idea is to choose a function $\ell$ which behaves arbitrarily at many separated scales, thus yielding a rich class $\mathcal{F}$. However since $f_*$ takes $n$ values in $[-1, 1]$ which are inevitably packed closely together, we cannot hope for $\ell \circ f_*$ to mimic arbitrary functions. However it will be enough to

choose $\ell$ so that $\ell \circ \mathcal{F}$ includes arbitrary functions which are *slowly varying*. Here and throughout the remainder of this section, we set

$$k = n^{1/1000}, \tag{27}$$

$$\lambda = \frac{\sqrt{\log k}}{C\eta\sqrt{n}} \tag{28}$$

where $C$ is a sufficiently large constant to be chosen later (and $\eta$ is small depending on $C$).

**Definition E.2.** The function $f : X \to [-1, 1]$ is **slowly varying** if for all $1 \leq i \leq n - 1$,

$$|f(x_{i+1}) - f(x_i)| \leq \frac{1}{n^{1.02}}. \tag{29}$$

Further, $f : X \to [-\lambda, \lambda]$ is **good** if it is slowly varying and satisfies $|f(x_i)| = \lambda$ for all $i$ such that

$$n^{-1/10} \leq n^{-9/10}i - \lfloor n^{-9/10}i \rfloor \leq 1 - n^{-1/10}. \tag{30}$$

Informally a good function is a $\{\pm\lambda\}$-valued function which takes constant values on each of $n^{1/10}$ intervals, except that near the boundaries it can change somewhat gradually between $\lambda$ and $-\lambda$. We next show that any $\{\pm\lambda\}$-valued function agrees with some good function at all $i$ satisfying equation 30. Define a partition

$$[n] = \bigcup_{m=1}^{M} P_m \tag{31}$$

as follows. $P_m$ consists of all $i \in [n]$ for which $\lfloor n^{-9/10}i \rfloor = m$ and equation 30 holds, except for the final part $P_M$ which consists of $i$ for which equation 30 does not hold. Note that $n^{9/10}/4 \leq |P_M| \leq n^{9/10}$, and for $m < M$ we have $|P_m| \leq n^{9/10}$, with $|P_m| \geq n^{9/10}/2$ for all but at most 1 value of $m < M$.

**Proposition E.3.** *For any function $g : [M - 1] \to \{\pm\lambda\}$ there exists a good function $f$ such that*

$$f(x_i) = g(m)$$

*for all $i \in P_m$ with $m < M$.*

*Proof.* The only thing to check is that $f$ can be extended to $P_M$ while obeying equation 29. This is clear because each interior interval in $P_M$ has length at least $n^{4/5} \gg \lambda n^{1.02}$. $\square$

Next we show that choosing $\ell$ appropriately allows $\ell \circ \mathcal{F}$ to contain any desired collection of good functions. This is the only place where the precise definition of $f_*$ is important; we need to be careful that the $n^{-1.01}$ spacing between adjacent values gives enough freedom for 1-Lipschitz $\ell$.

**Proposition E.4.** *Let $k, \lambda, f_*, \mathcal{F}$ be as above. For any good functions $f_1, \ldots, f_k : X \to [-\lambda, \lambda]$, there is a 1-Lipschitz function $\ell : \mathbb{R} \to \mathbb{R}$ such that $\ell \circ \mathcal{F}$ contains all functions $f_1, \ldots, f_k$ as well as the zero function.*

*Proof.* Let $\alpha = 1 - 100n^{-0.01}$. We require $\ell$ to take the values

$$\ell(\alpha^j f_*(x_i)) = f_j(x_i)$$

for all $1 \leq j \leq k$, and $\ell(0) = 0$. If this is possible, then by definition $\ell \circ \mathcal{F}$ is as required: take $t_j = \alpha^j$ for $\ell(t_j f_*(x)) = f_j(x)$, and $t = 0$ for the zero function. It remains to prove that $\ell$ is 1-Lipschitz on this finite set of inputs so a 1-Lipschitz extension to $\mathbb{R}$ exists.

For $j \neq j' \in [k]$ and any $i, i' \in [n]$, we find

$$|\alpha^j f_*(x_i) - \alpha^{j'} f_*(x_{i'})| \geq \alpha^k (f_*(x_n) - \alpha) \geq n^{-1/10} \geq 2\lambda \geq |f_j(x_i) - f_{j'}(x_{i'})|.$$

For $j = j'$ but $i \neq i'$ we similarly find

$$|\alpha^j f_*(x_i) - \alpha^j f_*(x_{i'})| \geq \frac{\alpha^k |i - i'|}{n^{1.01}} \geq \frac{|i - i'|}{n^{1.02}} \geq |f_j(x_i) - f_j(x_{i'})|,$$

where the first inequality is immediate from the construction, the second follows from equation 27, and the third from equation 29. Finally $\ell(0) = 0$ is no problem since $\alpha^j f_*(x_i) \geq n^{-1/10} \geq \lambda$ for all $i, j$. $\square$

We have just shown that it is possible to construct (by suitable choice of $\ell$) many arbitrary good functions at the appropriate scale $\lambda$. Next we argue that constructing these functions suffices for the desired counterexample. We note that $\lambda$ was carefully chosen so that the main term equation 32 is maximized. Moreover it is important that the error term equation 33 is much smaller, which holds because $|P_M|$ is smaller than $n$ by a power of $n$ factor. (This is not clear from the statement below, but is used in equation 35.)

**Lemma E.5.** *Recall the partition equation 31. There exist functions $g_1, \ldots, g_k : [M] \to \{\pm\lambda\}$ such that with $g_0$ the zero function,*

$$\frac{1}{n}\mathbb{E}_{\vec{\epsilon}} \max_{0 \leq j \leq k} \sum_{m=1}^{M} \sum_{i \in P_m} \epsilon_i g_j(m) - \eta g_j(m)^2 \geq \frac{\log k}{C^{1.5} n \eta}. \tag{32}$$

*Moreover for any such functions and any additional $g_* : P_M \to [-\lambda, \lambda]$,*

$$\frac{1}{n}\mathbb{E}_{\vec{\epsilon}} \max_{0 \leq j \leq k} \left| \sum_{i \in P_M} \left( \epsilon_i g_j(m) - \eta g_j(m)^2 \right) - \left( \epsilon_i g_*(x_i) - \eta g_*(x_i)^2 \right) \right| \leq \frac{\log k}{C^{1.9} n \eta}. \tag{33}$$

*Proof.* To show equation 32, we note first that deterministically,

$$\frac{1}{n}\sum_{m=1}^{M} \sum_{i \in P_m} \eta g_j(m)^2 = \eta \lambda^2 \stackrel{equation\ 28}{\leq} \frac{\log k}{C^2 n \eta}. \tag{34}$$

We fix any $\vec{\epsilon} \in \{\pm 1\}^n$ on which at least $M/10$ of the $m \in [M]$ are *friendly* (we write $m \in F \subseteq [M]$), meaning $S_m \equiv \left| \sum_{i \in P_m} \epsilon_i \right|$ satisfies

$$n^{9/20}/100 \leq S_m \leq 100 n^{9/20}.$$

Recalling the discussion below equation 31, it is easy to see $\vec{\epsilon}$ satisfies this with probability at least $1/2$. For any such $\vec{\epsilon}$ we choose the functions $g_i$ randomly; each is uniform over all $2^M$ functions $g_i : [M] \to \{\pm\lambda\}$. Then for each $1 \leq j \leq k$ we have that

$$\frac{1}{n}\sum_{m=1}^{M} \sum_{i \in P_m} \epsilon_i g_j(m) = \frac{1}{n}\sum_{m=1}^{M} S_m g_j(m) = \frac{1}{n}\left( \sum_{m \in F} S_m g_j(m) + \sum_{m \notin F} S_m g_j(m) \right).$$

Conditionally on $\vec{\epsilon}$, for each $1 \leq j \leq k$ the sum $\sum_{m \in F} S_m g_j(m)$ is very close to a Gaussian with variance $\Theta(n)$. In particular the Berry–Esseen theorem shows this sum, normalized by its standard deviation, has Kolmogorov–Smirnov distance from standard Gaussian at most $n^{-\Omega(1)}$. Meanwhile the sum over $m \notin F$ is non-negative with probability at least $1/2$. So for small constant $c_1$,

$$\mathbb{P}\left[ \frac{1}{n}\sum_{m=1}^{M} \sum_{i \in P_m} \epsilon_i g_j(m) \geq \lambda \cdot \frac{c_1 \sqrt{\log k}}{\sqrt{n}} \,\middle|\, \vec{\epsilon} \right] \geq 10/k.$$

By equation 28, choosing $C$ large depending on $c_1$ above, we find

$$\lambda \cdot \frac{c_1 \sqrt{\log k}}{\sqrt{n}} \geq \frac{\log k}{C^{1.1} n \eta}.$$

Comparing with equation 34, we find that with constant probability, the left-hand side of equation 32 (inside the expectation) is at least $\frac{\log k}{C^2 n \eta}$. Since $g_0 \equiv 0$, the left-hand side of equation 32 is always non-negative, and so equation 32 holds.

Continuing to equation 33, similarly to equation 34 the squared terms are deterministically bounded. The remaining terms are even more negligible since a sub-gaussian tail bound implies

$$\frac{1}{n}\mathbb{E}_{\vec{\epsilon} \in \{\pm 1\}^M} \max_{0 \leq j \leq k} \left| \sum_{i \in P_M} \left( \epsilon_i (g_j(m) - g_*(x_i)) \right) \right| \leq \lambda n^{-1} \sqrt{|P_M|} \log^5(n) \leq n^{-1.01}. \tag{35}$$

This completes the proof. $\square$

We are now ready to apply the preceding lemmata to conclude the proof of the main result.

*Proof of Theorem 2.6.* We first show equation 9. Fix $\mathbf{x} = (x_1, \ldots, x_n)$ and let $k, \lambda, f_*, \mathcal{F}$ be as in equation 27, equation 28, equation 26, equation 11. By Proposition E.1,

$$R_n^\eta(\mathcal{F}) \leq \frac{1}{2n\eta}. \tag{36}$$

We consider the partition in equation 31 of $[n]$ and apply Lemma E.5 to find functions $g_1, \ldots, g_k : [M] \to \{\pm\lambda\}$. By Proposition E.3, for each $1 \leq j \leq k$ there is a good function $f_j$ such that $f_j(x_i) = g_j(m)$ for all $i \in P_m$ where $m < M$. We let $f_0$ be the zero function (which similarly extends $g_0$). Using equation 33 to control the error on $P_M$, we find using equation 32 that

$$\frac{1}{n}\mathbb{E}_{\vec{\epsilon}} \max_{0 \leq j \leq k} \sum_{i=1}^n \epsilon_i f_j(x_i) - \eta f_j(x_i)^2 \geq \frac{\log k}{C^2 n\eta}. \tag{37}$$

We finally choose $\ell$ as in Proposition E.4 using the functions $f_0, f_1, \ldots, f_k$ as above. This guarantees that $\ell \circ \mathcal{F}$ contains all $f_j$ for $0 \leq j \leq k$. Recalling equation 36 and equation 37, we conclude that

$$\mathcal{R}_n^\eta(\ell \circ \mathcal{F}) \geq \frac{\log k}{C^2 n\eta} \geq \Omega(\log n) \cdot \mathcal{R}_n^\eta(\mathcal{F}).$$

We next turn to equation 10. The proof is almost exactly the same; just take each $x_i$ to be IID uniform on the fixed set $\{x_1, \ldots, x_n\}$ just considered. Elementary Chernoff estimates imply that with probability $1 - n^{-100}$, at most $n^{0.95}$ of the $n$ indices lie in $P_M$. Moreover with the same probability, for all $P_m$ with $|P_m| \geq n^{9/10}/2$ (i.e. all but at most 1), the number of IID indices lying in $P_m$ will be $|P_m| \cdot (1 \pm n^{-\Omega(1)})$. Then exactly the same proof as above works, e.g. the use of Berry-Esseen in proving Lemma E.5 is still valid. $\square$

