# OpenReview forum: "On Contraction of Sequential and Offset Rademacher Complexities"
_ICML.cc/2026/Conference — ICML 2026 regular_

### Official Review · Reviewer_3pZw · 2026-02-20

**Soundness:** 3
**Presentation:** 4
**Significance:** 2
**Originality:** 3
**Overall Recommendation:** 5
**Confidence:** 3

**Summary:**

This paper extend the  Ledoux–Talagrand contraction result to variants of the Rademacher complexity.

**Compliance With Llm Reviewing Policy:**

Affirmed.

**Key Questions For Authors:**

Can you include a short section on the significance of these results for empiricists?

**Limitations:**

A discussion of the significance of these results to deep learning models, and the limitations therein would be helpful to the broader audience.

A thorough verification of the proofs requires careful reading of the appendices.

**Strengths And Weaknesses:**

The paper is well written, although the material is accessible only to the specialist.   The results are qualitatively tight within polylog factors.

---

> ### Author Rebuttal · Authors · 2026-03-26
>
> We thank the reviewer for their careful attention to our work.
>
> > Can you include a short section on the significance of these results for empiricists?
>
> We view our work as being primarily theoretical in nature, with the significance for empiricists being that our results help develop intuition for interactive problems and what kinds of problems are solvable in these settings and what kinds are not.

---

> > ### Author Rebuttal · Reviewer_3pZw · 2026-03-31
> >
> > Blve a short section on the practical implications would increase the impact of this paper.

---

### Official Review · Reviewer_843K · 2026-02-25

**Soundness:** 3
**Presentation:** 2
**Significance:** 3
**Originality:** 3
**Overall Recommendation:** 5
**Confidence:** 2

**Summary:**

This paper studies the Rademacher complexity in statistical learning theory. More precisely, the authors extend the Ledoux-Talagrand contraction lemma to sequential and offset Rademacher complexities. Indeed, while the lemma was developed for classical Rademacher complexity, its application to sequential and offset variants has been underexplored. For sequential Rademacher complexity, the authors extend the analysis to tree-dependent settings, moving beyond the worst-case analysis previously established. They also consider the data-dependent offset Rademacher complexity.
Additionally, they provide some lower bounds to show the tightness of their results.

**Compliance With Llm Reviewing Policy:**

Affirmed.

**Final Justification:**

The rebuttal addressed my concerns.

**Key Questions For Authors:**

1. Related to the weakness "_Worst-case vs. data-dependent analysis._", when would the worst-case contraction lemma be tighter than the data-dependent contraction lemma?

2. About the weakness "_Theorem 2.6/2.3._", can you clarify why these assumptions are not necessary to analyze the tightness of the results?

**Limitations:**

yes

**Strengths And Weaknesses:**

**Strengths.**

The results seem significant and might help future researchers to derive theoretical guarantees for online learning or regression. Moreover, the tightness of their results is justified with lower bounds and the necessity of several assumptions involved in the results is justified.

**Weaknesses.**

- _Worst-case vs. data-dependent analysis._ While the paper presents contraction lemmas for the worst-case scenario, it does not compare when the worst-case contraction lemma yields tighter bounds than its data-dependent counterpart. Specifically, the conditions under which the data-dependent lemmas outperform the worst-case ones remain unaddressed.

- _Paper's readability._ The paper's readability could be improved:

    - _Section 1.1._ The presentation of function class size notions is dense; some concepts defined here could be introduced later (as some are used only in the proofs).

    - _Notation $\mathcal{R}\_n^{\eta}(\mathcal{F})$._ The notation $\mathcal{R}\_n^{\eta}(\mathcal{F})$ is ambiguous: the paper uses it for data-dependent Rademacher complexity, but one might reasonably interpret it as worst-case complexity.

    - _Theorem 2.2/2.3._ The meaning of Theorems 2.2 and 2.3 is hard to parse; some explanations before or after the theorems would have been welcome.

    - _Theorem 2.4._ The statement of Theorem 2.4 is unclear: does the constraction $\phi$ needs to be Lipschitz for the result to hold? Moreover, the notation $\lesssim$ is not defined.

    - _Theorem 2.6/2.3._ An assumption is missing in Theorem 2.6 to justify the tightness of Theorem 2.4's contraction (Donsker function). Also, Theorem 2.3 omits the assumption that the hypothesis class is C-integral to justify the necessity of several factors in Theorem 2.1. Without these assumptions, we do not know if the conclusions still hold (i.e., the necessity of at least a factor $\Omega(\log n)$ in Theorem 2.4 and the necessity of the littlestone dimension and polylogarithmic factor in $n$ for Theorem 2.1).

**(Minor) Remarks.**

- _Line 37-right col._ Writing "1-Lipschitz function" (instead of "Lipschitz function") would better clarify that Eq. (1) depends on the Lipschitz constant.

- _Line 135-left col._ The notation $\epsilon$ might confuse the reader as it is used for both the scale parameter and Rademacher variables.

- _Line 235-right col._: The phrase "One setting where Theorem 2.5 can be applied..." should reference Theorem 2.4 instead.

---

> ### Author Rebuttal · Authors · 2026-03-26
>
> We thank the reviewer for their careful attention to our work.  We respond to each point individually below.
>
> > Worst-case vs. data-dependent analysis. While the paper presents contraction lemmas for the worst-case scenario, it does not compare when the worst-case contraction lemma yields tighter bounds than its data-dependent counterpart. Specifically, the conditions under which the data-dependent lemmas outperform the worst-case ones remain unaddressed.
>
> Thank you for pointing this out; we will better clarify this in the revision. To summarize, prior work demonstrated that worst-case sequential Rademacher complexity does satisfy contraction up to a polylogarithmic in
>  factor. Note that this is much, much tighter than what can be achieved with tree-dependent sequential Rademacher complexity, as our work demonstrates. In some function classes, this polylogarithmic factor in contraction of worst-case Rademacher complexity can be replaced by a constant factor; again, this cannot be achieved for tree-dependent complexity. Prior work does not analyze contraction for offset Rademacher complexity (worst-case or not).
>
> Thus our work suggests that one can only reasonably expect sequential Rademacher complexity of a function class composed with a Lipschitz function to be comparable to that of just the function class itself in a *worst-case* sense.
>
>
> > Paper's readability. The paper's readability could be improved
>
> Thank you for these suggestions, we will endeavor to improve the readability in the revision.
>
>
> > Related to the weakness "Worst-case vs. data-dependent analysis.", when would the worst-case contraction lemma be tighter than the data-dependent contraction lemma?
>
> To clarify, in some sense these are fundamentally incomparable as the worst-case contraction inequality only applies to worst-case complexities, which can be substantially larger than tree-dependent ones.  On the other hand, our results show that one cannot hope in general to take advantage of this fact for the purposes of contraction, because we provably cannot obtain contraction with a small constant unless we consider worst-case contraction inequalities.
>
> > About the weakness "Theorem 2.6/2.3.", can you clarify why these assumptions are not necessary to analyze the tightness of the results?
>
> I think there may be some confusion on the part of the reviewer.  Theorem 2.3 discusses *sequential* Rademacher complexity, while Theorem 2.6 involves *offset* Rademacher complexity.  In particular, C-integrality is used in the sequential setting, but is not necessary in the offset setting.  We are happy to address any further questions related to this matter if it remains unclear.

---

> > ### Author Rebuttal · Reviewer_843K · 2026-04-02
> >
> > I am convinced by the rebuttal; thank you for clarifying my concerns. I have therefore raised my overall recommendation to "Accept".

---

### Official Review · Reviewer_jmE1 · 2026-03-11

**Soundness:** 4
**Presentation:** 4
**Significance:** 3
**Originality:** 4
**Overall Recommendation:** 5
**Confidence:** 2

**Summary:**

This work studies contraction properties of sequential and offset Rademacher complexities. The authors establish a contraction inequality with a polylogarithmic factor in $n$ for sequential Rademacher complexity of symmetric and integral function classes, and show that without the symmetry assumption the contraction factor must be at least $\sqrt{n}$. They also establish a contraction inequality with a polylogarithmic factor in $n$ for offset Rademacher complexity of symmetric, convex, and Donsker function classes. Furthermore, they show that without convexity the offset Rademacher complexity can be zero while the contracted analogue remains positive.

**Compliance With Llm Reviewing Policy:**

Affirmed.

**Key Questions For Authors:**

Could the authors elaborate on the following paragraph (which is also discussed in the introduction)?

> As mentioned previously, Theorem 2.1 can be used to construct algorithms with small regret through the relaxations approach of (Rakhlin et al., 2012); for example, if \(z\) is a tree with monotone paths (i.e. \(z_i \ge z_j\) for all \(i \ge j\)) and \(\mathcal{F}\) is the class of one-dimensional thresholds, then worst-case sequential Rademacher complexity does not decay (Littlestone, 1988), but Equation (6) can still lead to a nontrivial bound.

**Strengths And Weaknesses:**

This is a very interesting work which I believe will be appreciated by the community. Unfortunately, I do not have sufficient background in this area to provide constructive feedback on the technical contributions. I attempted to follow the proofs and only noticed a few minor issues:
- In Line 91, it is written Subsection 2. I think it was meant to refer either to Section 2 or Subsection 2.2.
- There are typos in Lines 742 and 751.

---

> ### Author Rebuttal · Authors · 2026-03-26
>
> We thank the reviewer for their careful attention to our paper.
>
> We also thank the reviewer for pointing out the typos and we will be sure to correct them in the revision.
>
> > Could the authors elaborate on the following paragraph (which is also discussed in the introduction)?
>
> Yes, we will clarify this point in the revision.  The key idea behind the cited relaxations approach is to use sequential Rademacher complexity on the future data as a potential to be used in solving a minimax problem; the solution to this minimax problem at each time step then becomes the prediction.  Critically, in the worst case, many otherwise simple function classes like thresholds in one dimension have infinite Littlestone dimension (a worst case notion), but may still be learnable for "realistic" or "easy" data sequences.  Our work potentially provides an avenue toward this goal through better understanding the structural properties of one of the common potentials used to construct these algorithms.

---

> > ### Author Rebuttal · Reviewer_jmE1 · 2026-04-02
> >
> > I have no further questions.

---

### Official Review · Reviewer_S1dc · 2026-03-13

**Soundness:** 3
**Presentation:** 3
**Significance:** 3
**Originality:** 3
**Overall Recommendation:** 4
**Confidence:** 4

**Summary:**

For the problem of generalizing the classical Ledoux-Talagrand contraction lemma to sequential and offset Rademacher complexities, this paper presents contraction results for tree-dependent sequential Rademacher complexity and offset Rademacher complexity, proves the necessity of the relevant assumptions, and analyzes the tightness of the bounds. The results have significant theoretical implications for the generalization analysis of machine learning.

**Compliance With Llm Reviewing Policy:**

Affirmed.

**Final Justification:**

My concerns have been partially resolved, and I will maintain my original positive score.

**Key Questions For Authors:**

Please refer to the Weaknesses for details.

**Limitations:**

Yes.

**Strengths And Weaknesses:**

**Strengths:**

1. The contraction results for sequential and offset Rademacher complexities are of great significance for understanding the sample complexity of composed function classes, and the contraction bounds proposed in this paper have potential application value in the generalization analysis of practical problem settings.

2. This paper not only provides upper bounds for the contraction property, but also verifies the necessity of assumptions such as symmetry and convexity, as well as the ineliminability of polylogarithmic factors and Littlestone dimension factors, through constructive proofs. The proof techniques are quite elegant.


**Weaknesses:**

1. While this paper reviews relevant work on sequential and offset Rademacher complexity, it does not quantitatively compare its results with existing contraction results for worst-case sequential and offset Rademacher complexity. For example, it does not analyze the difference in contraction factors between the tree-dependency case and the worst-case case, nor does it explain how its results improve or extend existing research, making the contribution of this paper unclear.

2. The contraction property of offset Rademacher complexity in Theorem 2.4 depends on the function class being a Donsker class. However, this paper only points out that empirical risk minimization cannot achieve a minimax rate for non-Donsker classes, without further analyzing the limitations imposed by the Donsker class assumption on the theoretical results. In practice, many nonparametric function classes (such as some deep neural network classes) are not Donsker classes. The possibility of extending the derived results to these function classes should be further explained, and the potential directions of weakening the Donsker assumption should be further discussed. Otherwise, the applicability of the theoretical results in this paper will be limited.

3. Can the quantitative factors of the contraction bound for sequential Rademacher complexity be further improved? The contraction bound given in Theorem 2.1 includes the 3/2 power of the Littlestone dimension and the 5/2 power of the logarithmic term. This paper only proves that the Littlestone dimension and the logarithmic term factor cannot be removed (with lower bounds), but does not analyze whether these factors are tight (e.g., whether the power of the Littlestone dimension can be improved from 3/2 to 1, or whether the order of the logarithmic factor can be reduced through more refined martingale analysis), nor does it provide examples of specific function classes to verify the tightness of bounds, making the quantitative guidance of the results insufficient.

---

> ### Author Rebuttal · Authors · 2026-03-26
>
> We thank the reviewer for their careful attention to our work.  We outline a few key points.
>
> > While this paper reviews relevant work on sequential and offset Rademacher complexity, it does not quantitatively compare its results with existing contraction results for worst-case sequential and offset Rademacher complexity.
>
> Thank you for pointing this out; we will better clarify this in the revision.  To summarize, prior work demonstrated that worst-case sequential Rademacher complexity *does* satisfy contraction up to a polylogarithmic in $n$ factor.  Note that this is much, much tighter than what can be achieved with tree-dependent sequential Rademacher complexity, as our work demonstrates.  In some function classes, this polylogarithmic factor in contraction of worst-case Rademacher complexity can be replaced by a constant factor; again, this cannot be achieved for tree-dependent complexity.  Prior work does not analyze contraction for offset Rademacher complexity (worst-case or not).
>
>
> > The contraction property of offset Rademacher complexity in Theorem 2.4 depends on the function class being a Donsker class.
>
> We agree that the Donsker assumption is limiting, but unfortunately our proof makes essential use of the Donsker property (in particular that single-scale discretization and multi-scale "chaining" bounds are of the same order); if we were to extend our analysis beyond the Donsker case we would suffer polynomial factors.  We agree that it is an interesting question for future research, to do this.
>
> > Can the quantitative factors of the contraction bound for sequential Rademacher complexity be further improved?
>
> These factors are likely not tight, and improving them would be an interesting direction for future research.  The goal of our paper was to understand the more qualitative properties that govern the extent to which contraction holds.

---

> > ### Author Rebuttal · Reviewer_S1dc · 2026-04-04
> >
> > The authors' rebuttal partially addressed my concerns, and I will maintain my rating.

---

### Decision · Program_Chairs · 2026-04-30

**Decision:**

Accept (regular)

**Comment:**

This paper studies sequential and offset Rademacher complexities, and develops contraction inequalities for Rademacher complexities of composite function classes under symmetry/convexity assumptions. All the reviewers agree that the paper gives a strong contribution to the machine learning theory as Rademacher complexity is a fundamental tool. Especially, reviewers appreciate the lower bounds and the justification on the necessity of the assumptions. The authors also give some suggestions on the presentation and practical implications. I would encourage the authors to incorporate these suggestions in the revision.